

# Phylogeny of *Crataegus* (Rosaceae) based on 257 nuclear loci and chloroplast genomes: evaluating the impact of hybridization

Aaron Liston[1,*], Kevin A. Weitemier[1,2], Lucas Letelier[1], János Podani[3], Yu Zong[1,4], Lang Liu[5] and Timothy A. Dickinson[6,7,*]

[1] Department of Botany and Plant Pathology, Oregon State University, Corvallis, OR, United States of America
[2] Department of Fisheries and Wildlife, Oregon State University, Corvallis, OR, United States of America
[3] Department of Plant Systematics, Ecology and Theoretical Biology, Eötvös Lorand University, Budapest, Hungary
[4] College of Chemistry & Life Sciences, Zhejiang Normal University, Jinhua, Zhejiang, China
[5] Department of Cell and Systems Biology, University of Toronto, Toronto, Ontario, Canada
[6] Department of Natural History, Royal Ontario Museum, Toronto, Ontario, Canada
[7] Department of Ecology and Evolutionary Biology, University of Toronto, Toronto, Ontario, Canada
[*] These authors contributed equally to this work.

Corresponding author
Aaron Liston,
aaron.liston@oregonstate.edu

## ABSTRACT

**Background**. Hawthorn species (*Crataegus* L.; Rosaceae tribe Maleae) form a well-defined clade comprising five subgeneric groups readily distinguished using either molecular or morphological data. While multiple subsidiary groups (taxonomic sections, series) are recognized within some subgenera, the number of and relationships among species in these groups are subject to disagreement. Gametophytic apomixis and polyploidy are prevalent in the genus, and disagreement concerns whether and how apomictic genotypes should be recognized taxonomically. Recent studies suggest that many polyploids arise from hybridization between members of different infrageneric groups.

**Methods**. We used target capture and high throughput sequencing to obtain nucleotide sequences for 257 nuclear loci and nearly complete chloroplast genomes from a sample of hawthorns representing all five currently recognized subgenera. Our sample is structured to include two examples of intersubgeneric hybrids and their putative diploid and tetraploid parents. We queried the alignment of nuclear loci directly for evidence of hybridization, and compared individual gene trees with each other, and with both the maximum likelihood plastome tree and the nuclear concatenated and multilocus coalescent-based trees. Tree comparisons provided a promising, if challenging (because of the number of comparisons involved) method for visualizing variation in tree topology. We found it useful to deploy comparisons based not only on tree-tree distances but also on a metric of tree-tree concordance that uses extrinsic information about the relatedness of the terminals in comparing tree topologies.

**Results**. We obtained well-supported phylogenies from plastome sequences and from a minimum of 244 low copy-number nuclear loci. These are consistent with a previous morphology-based subgeneric classification of the genus. Despite the high heterogeneity of individual gene trees, we corroborate earlier evidence for the importance of hybridization in the evolution of *Crataegus*. Hybridization between

subgenus *Americanae* and subgenus *Sanguineae* was documented for the origin of *Sanguineae* tetraploids, but not for a tetraploid *Americanae* species. This is also the first application of target capture probes designed with apple genome sequence. We successfully assembled 95% of 257 loci in *Crataegus*, indicating their potential utility across the genera of the apple tribe.

## INTRODUCTION

The relative importance of hybridization in the evolution of *Crataegus* L. (and in some other Rosaceae) has been contentious in the past, and is reviewed elsewhere (*Dickinson, 2018*); it suffices for now to note that just within subtribe Malinae (fleshy fruits derived from hypanthial, or inferior, ovaries; Rosaceae subfamily Amygdaloideae) there are molecular data to document hybridization in several large genera (*Burgess et al., 2015*; *Cushman et al., 2017*; *Hamston et al., 2018*; *Li et al., 2014*; *Li et al., 2017*; *Liu et al., 2020*; *Németh et al., 2020*), including *Crataegus*. This genus of approximately 200 or more species (*Phipps, 2015*) is found in a small clade of five genera and ca. 270 species in total that is sister group to the remaining 24 genera and ca. 530 species of Malinae (*Mabberley, 2008*; *Campbell et al., 2015*; *Tropicos.org, 2021*). Together with *Hesperomeles* Lindl. (*Li et al., 2012*; *Liu et al., 2020*), *Crataegus* diverged from unarmed, berry-fruited *Amelanchier*, *Malacomeles*, and *Peraphyllum* by the acquisition of thorns and polypyrenous drupes whose pyrenes enclose a single seed (*Campbell et al., 2007*; *Lo & Donoghue, 2012*; *Potter et al., 2007*; *Xiang et al., 2017*; *Zhang et al., 2017*). *Crataegus* encompasses considerable variation in its thorns, leaves, flowers, and fruits, such that there is a well-developed infrageneric classification (Table 1; *Loudon, 1838*; *Palmer, 1925*; *Phipps, 2015*; *Schneider, 1906*; *Ufimov & Dickinson, 2020*) that is supported by DNA sequence data (Fig. 1; *Lo & Donoghue, 2012*; *Lo et al., 2009a*; *Zarrei et al., 2015*; *Ufimov et al., 2021*). Nevertheless, comparisons of microsatellites, and of nuclear and chloroplast loci (*Lo, Stefanović & Dickinson, 2009b*; *Lo, Stefanović & Dickinson, 2010*), and ribosomal DNA (ITS2) copy number variation correlated with differences in ploidy level (*Zarrei, Stefanović & Dickinson, 2014*), strongly suggest that hybridization between infrageneric groups (subgenera, sections, series) has played an important role in the diversification of *Crataegus*. Prior to these data becoming available, however, with one exception (*Phipps, 1988*) hybridization was not seen to be a factor in *Crataegus* diversification (*Haines, 2011*; *Phipps, 2005*), and several new species were described in North America during the period 1980–2007 with their possible hybrid origin being either ignored or explicitly rejected. Subsequently, however, taxonomic and floristic works on *Crataegus* have reversed this trend (*Kurtto, Sennikov & Lampinen, 2013*; *Lance, 2014*; *Phipps, 2013*; *Phipps, 2015*). Because both ploidy level variation and hybridization are usually associated with uniparental reproduction by means of gametophytic apomixis the cumulative consequences, in terms of taxonomic complexity, have been considerable.

**Table 1** **Hawthorn individuals used here as sources of leaf tissue for DNA extraction, and for which ploidy level determined previously by flow cytometry (publications cited).** Classification follows *Lo, Stefanović & Dickinson (2007)*; *C. germanica*, *Ufimov (2013)*; *C.* subg. *Sanguineae*, *Phipps (2015)*, and *Ufimov & Dickinson, (2020)*. Voucher specimens are deposited in the Green Plant Herbarium of the Royal Ontario Museum (TRT). TRT accession numbers are linked to online specimen images; Target capture sequence data will be deposited in the National Center for Biotechnology Information Sequence Read Archive (NCBI SRA); TADCR numbers are searchable on the Barcode of Life Data System (BOLD; http://v4.boldsystems.org/). Sporophytic chromosome numbers ($2n$) are reported as multiples of the base number, $x = 17$, based on flow cytometric determinations. Stamen numbers per flower ($A_\#$). Localities represented here are in the United States or Canada. One sample of *C. chrysocarpa* lacks a flow cytometric ploidy determination; it is presumed $4x$ based on data from another indistinguishable individual in the same population.

| | TRT Accession, NCBI SRA, BOLD, and sample numbers (this study) | $2n$; $A_\#$ | Collector & number | Publication | State level | County level | Locality; Latitude, Longitude (degrees) or accession number for botanical garden specimens |
|---|---|---|---|---|---|---|---|
| *Crataegus* L. | | | | | | | |
| subg. *Mespilus* Ufimov & T.A. Dickinson | | | | | | | |
| sect. *Mespilus* T. A. Dickinson & E. Y. Y. Lo | | | | | | | |
| C. germanica (L.) Kuntze | TRT00026642; SAMN16630157; TADCR097-10; s07 | $2x$ $A_{30}$ | Dickinson, T.A. s.n. | *Zarrei et al. (2015)* | California | Alameda Co. | Cultivated; U. of California Botanic Garden (78.0184) |
| subg. *Brevispinae* (Beadle) Ufimov & T. A. Dickinson | | | | | | | |
| sect. *Brevispinae* Beadle ex Schneider | | | | | | | |
| C. brachyacantha Sarg. & Engelm. | TRT00000028; SAMN16630160; TADCR073-10; s10 | $2x$ $A_{20}$ | Reid, C. 5203 | *Talent & Dickinson (2005)*; *Zarrei et al., (2015)* | Louisiana | Morehouse Parish | ca. 3.75 miles NE of Oak Ridge; 32.66, -91.73 |
| subg. *Crataegus* | | | | | | | |
| sect. *Crataegus* | | | | | | | |
| ser. *Crataegus* | | | | | | | |
| C. monogyna Jacq. | TRT00000394; SAMN16630171; TADCR109-10; s21 | $2x$ $A_{20}$ | Dickinson, T.A. 2003-79 | *Zarrei et al. (2015)* | Ontario | Middlesex Co. | Denfield sideroad 0.25 miles S of Hwy 16; 43.07, -81.40 |

**Table 1** (*continued*)

| | TRT Accession, NCBI SRA, BOLD, and sample numbers (this study) | $2n$; $A_\#$ | Collector & number | Publication | State level | County level | Locality; Latitude, Longitude (degrees) or accession number for botanical garden specimens |
|---|---|---|---|---|---|---|---|
| subg. *Americanae* El-Gazzar | | | | | | | |
| sect. *Coccineae* Loudon | | | | | | | |
| ser. *Aestivales* (Sarg.) Rehder | | | | | | | |
| *C. opaca* Hook. & Arn. | TRT00002042; SAMN16630158; TADCR020-10; s08 | $2x$ $A_{20}$ | Dickinson, T.A. 2003-33 | *Talent & Dickinson (2005)*; *Zarrei et al., (2015)* | Louisiana | De Soto Parish | Cultivated; Trey Lewis home place, 31.84, -93.77 |
| ser. *Crus-galli* (Loud.) Rehder | | | | | | | |
| *C. crus-galli* L. | TRT00002636; SAMN16630159; s09 | $2x$ $A_{20}$ | Talent, N. NT489 | Published here with permission of N. Talent | Georgia | Houston Co. | South of Big Indian Creek, road verge; 32.41, -83.57 |
| ser. *Punctatae* (Loud.) Rehder | | | | | | | |
| *C. punctata* Jacq. | TRT00002247; SAMN16630155; TADCR104; s05 | $2x$ $A_{20}$ | Purich, M.A. 81 | *Zarrei et al. (2015)* | Ontario | Durham R.M. | Bowmanville, between two forks of Bowmanville Creek; 43.90, -78.68 |
| ser. *Rotundifoliae* (Egglest. ex Egglest.) Rehder | | | | | | | |
| *C. chrysocarpa* Ashe | TRT00000270; SAMN16630156; s06 | $4x$ $A_{10}$ | Lo, E.Y.Y. EL-122 | *Talent & Dickinson (2005)* | Idaho | Nez Perce Co. | Hwy 3, at Little Potlatch Creek; 46.52, -116.73 |
| *C. chrysocarpa* Ashe | TRT00020434; SAMN16630167; s17 | $A_{10}$ | Coughlan, J. JC174 | See TRT barcode link. | Washington | Okanogan Co. | N side of Palmer Lake; 48.92 -119.64 |
| ser. *Triflorae* (Beadle) Rehder | | | | | | | |
| *C. triflora* Chapm. | TRT00021429; SAMN16630172; TADCR107; s22 | $2x$ $A_{30}$ | Dickinson, T.A. 2003-23 | *Talent & Dickinson (2005)*; *Zarrei et al., (2015)* | Alabama | Autauga Co. | Jones Bluff, SSW of Peace; 32.40, -86.78 |

**Table 1** (*continued*)

| | TRT Accession, NCBI SRA, BOLD, and sample numbers (this study) | 2*n*; A# | Collector & number | Publication | State level | County level | Locality; Latitude, Longitude (degrees) or accession number for botanical garden specimens |
|---|---|---|---|---|---|---|---|
| sect. *Macracanthae* Loudon | | | | | | | |
| ser. *Macracanthae* (Loud.) Rehder | | | | | | | |
|   *C. calpodendron* (Ehrh.) Medikus | TRT00000105; SAMN16630151; s01 | 2*x* A$_{20}$ | Dickinson, T.A. 2002-07A | *Talent & Dickinson (2005)* | Massachusetts | Suffolk Co. | Cultivated; Arnold Arboretum (AA277-68A) |
|   *C. macracantha* Lodd. ex Loud. var. *occidentalis* (Britt.) Egglest. | TRT00000142; SAMN16630161; s11 | 4*x* A$_{10}$ | Talent, N. NT347 | Published here with permission of N. Talent | Colorado | Boulder Co. | Gregory Canyon; 40.00, -105.29 |
|   *C. macracantha* Lodd. ex Loud. var. *occidentalis* (Britt.) Egglest. | TRT00020260; SAMN16630168; TADCR280; s18 | 4*x* A$_{10}$ | Coughlan, J. JC168 | *Zarrei et al. (2015)* | Washington | Okanogan Co. | Eastside River Rd., N of Omak; 48.50, -119.50 |
| subg. *Sanguineae* Ufimov | | | | | | | |
| sect. *Salignae* T.A. Dickinson & Ufimov | | | | | | | |
| ser. *Cerrones* J.B. Phipps | | | | | | | |
|   *C. rivularis* Nutt. ex Torr. & A.Gray | TRT00000946; SAMN16630154; TADCR181; s04 | 4*x* A$_{10}$ | Lo, E. EL-199 | *Talent & Dickinson (2005)*; *Zarrei et al., (2015)* | Idaho | Bear Lake Co. | US 89 W of Whitman Hollow; 42.34, -111.21 |
|   *C. rivularis* Nutt. ex Torr. & A.Gray | TRT00000965; SAMN16630162; TADCR165; s12 | 4*x* A$_{10}$ | Dickinson, T.A. 2007-02 | *Zarrei et al. (2015)* | Nevada | Elko Co. | Starr Valley, on N side of Dennis Flats Road; 41.01, -115.27 |
|   *C. rivularis* Nutt. ex Torr. & A.Gray | TRT00000976; SAMN16630170; s20 | 4*x* A$_{10}$ | Talent, N. NT357 | Published here with permission of N. Talent | New Mexico | Rio Arriba Co. | US84, S end of Chama; 36.87, -106.58 |
|   *C. saligna* Greene | TRT00001025; SAMN16630163; s13 | 2*x* A$_{20}$ | Dickinson, T.A. 2001-07 | *Dickinson et al. (2008)* | Colorado | Rio Blanco Co. | Rio Blanco Rd 8, N bank of White River; 40.03, -107.86 |
|   *C. saligna* Greene | TRT00001047; SAMN16630164; TADCR120; s14 | 2*x* A$_{20}$ | Dickinson, T.A. 2004-05 | *Talent & Dickinson (2005)*; *Zarrei et al. (2015)* | Utah | Duchesne Co. | River Road, 4 miles N of Duchesne; 40.21, -110.41 |

**Table 1** (*continued*)

| | TRT Accession, NCBI SRA, BOLD, and sample numbers (this study) | 2*n*; A# | Collector & number | Publication | State level | County level | Locality; Latitude, Longitude (degrees) or accession number for botanical garden specimens |
|---|---|---|---|---|---|---|---|
| sect. *Douglasianae* (Rehder) C.K. Schneid. | | | | | | | |
| ser. *Douglasianae* (Loud.) Poletiko | | | | | | | |
| *C. douglasii* Lindl. | TRT00001145; SAMN16630153; TADCR001; s03 | 4*x* A$_{10}$ | Lo, E. EL-11 | *Zarrei et al. (2015)* | Ontario | Grey Co. | Keppel Twp., Colpoy's Range; 44.80, -81.00 |
| *C. douglasii* Lindl. | TRT00001279; SAMN16630169; TADCR177; s19 | 4*x* A$_{10}$ | Lo, E. EL-170 | *Zarrei et al. (2015)* | Idaho | Adams Co. | W bank of Goose Creek, S of Last Chance Campground; 44.99, -116.19 |
| *C. douglasii* Lindl. | TRT00020479; SAMN16630166; s16 | 3*x* A$_{10}$ | Coughlan, J. JC224 | See TRT barcode link. | Oregon | Union Co. | Hwy 203, SE of Union; 45.13 -117.71 |
| *C. suksdorfii* (Sarg.) Kruschke | TRT00020315; SAMN16630174; s24 | 2*x* A$_{20}$ | Coughlan, J. JC033 | *Coughlan (2012)* | California | Siskiyou Co. | Fay Lane, just W of Scott R.; 41.40, -122.84 |
| *C. suksdorfii* (Sarg.) Kruschke | TRT00001805; SAMN16630165; s15 | 2*x* A$_{20}$ | Zika, P.F. 18485 | *Zarrei et al. (2015)* | Washington | Clark Co. | ca. 1.5 air miles NNW of Ridgefield; 45.83, -122.75 |
| sect. *Sanguineae* Zabel ex C.K. Schneid. | | | | | | | |
| ser. *Nigrae* (Loudon) Russanov | | | | | | | |
| *C. nigra* Waldst. and Kit. | TRT00001999; SAMN16630173; TADCR025; s23 | 2*x* A$_{20}$ | Dickinson, T.A. 2318-50 | *Talent & Dickinson (2005)*; *Zarrei et al. (2015)* | Québec | | Cultivated; Jardin Botanique de Montréal, Arboretum (2318-50) |
| ser. *Sanguineae* (Zabel ex C.K. Schneid.) Rehder | | | | | | | |
| *C. wilsonii* Sarg. | TRT00002055; SAMN16630152; TADCR114-10; s02 | 2*x* A$_{20}$ | Dickinson, T.A. s.n. | *Talent & Dickinson, (2005)*; *Zarrei et al., (2015)* | Massachusetts | Suffolk Co. | Cultivated; Arnold Arboretum (AA749-74A) |
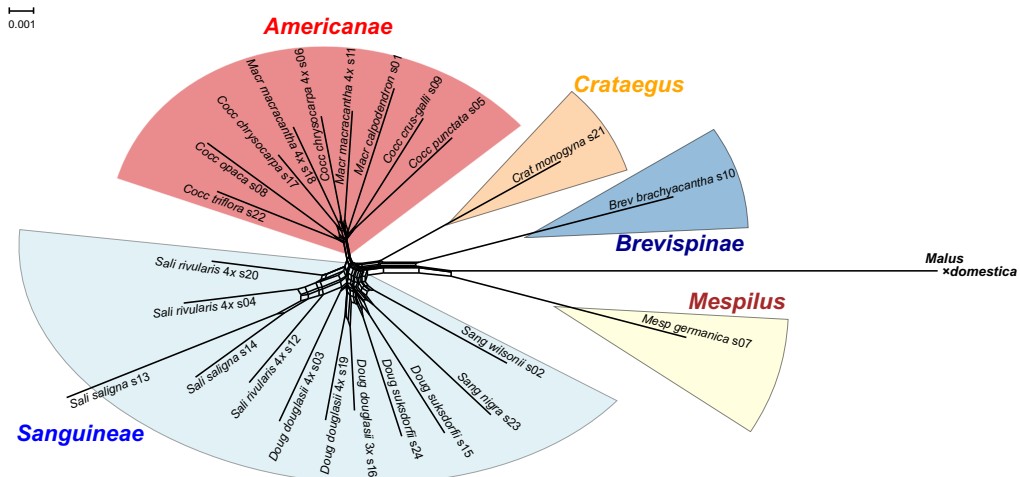

**Figure 1** **Splits network representation of 24 hawthorn individuals based on uncorrected p-distances between aligned sequences for 244 nuclear loci (529,827 positions).** Individuals are labeled with accession number (s00) + *species* + *Section (Aaaa)* as in Table 1. Labels also indicate ploidy level if tetraploid (4x, x = 17; otherwise diploid; Table 1). *Crataegus* subgenera shown in color. Network fit = 97%. Diagram produced with SplitsTree4 (*Huson & Bryant, 2005*) and ColorBrewer 2.0 (*Brewer, 2013*).

For an orthologous nucleotide site, its entire geneaological history can accurately be represented as a single evolutionary tree (*Ralph, Thornton & Kelleher, 2020*). However, once multiple nucleotides are examined, the processes of mutation and recombination will create different bifurcating histories among nucleotides in a genome. Once population divergence and speciation occur, incomplete lineage sorting (ILS) further contributes to discordant phylogenetic trees across the genome (*Degnan & Rosenberg, 2009*). Incomplete lineage sorting can be modeled and accounted for in methods that aim to reconstruct a species tree from an assemblage of individual gene trees (*Zhang et al., 2018*), and its potential impact on phylogenetic reconstruction can be quantified with gene and nucleotide site concordance measures (*Minh, Hahn & Lanfear, 2020a*). These methods require a large number of loci. Fortunately, current methods allow for the sequencing and efficient analysis of hundreds to thousands of nuclear loci (*Weitemier et al., 2014*; *Johnson et al., 2016*), and are applied here to estimate phylogenetic relationships among species of the genus *Crataegus*.

However, organismal diversification is not a strictly bifurcating process, due to the widespread occurrence of interspecific hybridization. Like recombination and ILS, hybridization will also result in conflict among gene trees, as each sequence obtained from an individual of a hybrid species will originate from only one of its progenitors. Thus, different gene trees will trace different ancestries, resulting in conflict. Discerning whether conflict results from ILS or reticulation is particularly difficult, but recent progress with methods based on nucleotide site pattern probabilities have made this tractable (*Blischak et al., 2018*), and this approach is applied here. Conflicts among nucleotide site patterns are also analyzed with a splits network (*Huson & Bryant, 2005*) to provide a visualization of reticulate events.

Alternatively, all alleles at a locus can be obtained for a hybrid individual (*Rothfels, 2021*), but the technical and analytical demands of this approach can be prohibitive, and may minimally impact phylogenetic reconstruction (*Kates et al., 2018*). Furthermore, the presence of multiple sequences per individual requires multi-labelled trees, which also present analytical challenges (*Rothfels, 2021*). For these reasons, we do not incorporate these methods.

In addition to using sequence evolution methods for inferring phylogenetic relationships and hybridization, we investigate the information available in the topologies of a collection of 257 gene trees. We use two related methods to do so. First, we employ phenetic descriptors of tree topology, singly and combined, using the Euclidean distance to compare trees in an ordination space (principal coordinates analysis). Second, we employ two related measures designed for situations in which the tips of the trees are related according to some extrinsic factor such as taxonomic group: the Related tree distance (RT), and a concordance measure. Both measures depend on collapsing a reference tree so as to represent only the relationships between extrinsic groups. In this way each of these measures (RT, rtCF) depicts the extent to which a given sample tree reflects the relationships seen in the reference tree between the extrinsic groups. In this way we seek to interrogate the structure of our gene trees for evidence of hybridization or other processes that would confound the ability of an individual gene tree to document relationships seen in the reference tree that we believe to have priority, based on non-molecular or other information. We use these tree descriptors to demonstrate their utility for analyses of the topological congruence among trees.

## MATERIALS & METHODS

### Taxon sampling

The focus of our work to date is *Crataegus* subg. *Sanguineae*, and the black-fruited hawthorns that are its North American representatives (*Coughlan et al., 2017*; *Coughlan, Stefanović & Dickinson, 2014*; *Dickinson et al., 1996*; *Dickinson et al., 2008*; *Dickinson & Love, 1997*; *Evans & Dickinson, 1996*; *Lo, Stefanović & Dickinson, 2009b*; *Lo, Stefanović & Dickinson, 2013*; *Ufimov & Dickinson, 2020*; *Zarrei, Stefanović & Dickinson, 2014*). We supplement these earlier results, where they were based on a limited number of loci, by obtaining total genomic DNA from leaf tissue dried on silica gel for 24 field- or garden-collected *Crataegus* individuals for which ploidy level had been determined previously by flow cytometry (Table 1; cf. *Talent & Dickinson, 2005*). Our sample encompasses all five *Crataegus* subgenera recognized at present, each one represented by one or more species shown to be diploid, for a total of 14 diploid *Crataegus* accessions (Table 1). These diploid accessions also include representatives of all the taxonomic sections of *C.* subg. *Americanae* and *C.* subg. *Sanguineae* (Table 1). To this sample we have added four accessions representing tetraploid species in *C.* subg. *Americanae*, *C. chrysocarpa* and *C. macracantha* (Table 1). We have also added six accessions representing two primarily tetraploid species in *C.* subg. *Sanguineae*, *C. douglasii* and *C. rivularis* (Table 1). One of the *C. douglasii* accessions is a triploid. Triploids are rare in *C. douglasii* (*Talent & Dickinson, 2005 Dickinson et al. 2021*). References below to tetraploids include this

triploid *C. douglasii* individual (s16, Table 1). In this way our sample comprises diploid and tetraploid representatives of each of subg. *Sanguineae* sections *Douglasianae* and *Salignae* (Table 1). The *Sanguineae* tetraploids have been shown to be allopolyploids that combine *Americanae* and *Sanguineae* genomes (Table 2; *Zarrei, Stefanović & Dickinson, 2014*). The *Americanae* tetraploids were chosen because they are the only widespread, common species of this subgenus with ranges that overlap at least in part with those of *C.* sections *Douglasianae* and *Salignae* (*Dickinson et al. 2021*; *Phipps, 2015*) and so are likely representative of the *Americanae* parent(s) of the intersubgeneric hyrids. The corresponding representatives of the maternal *Sanguineae* parents are the two diploid accessions of *C. suksdorfii* sensu lato and of *C. saligna* Greene (Table 1; cf. *Zarrei, Stefanović & Dickinson, 2014*). North American hawthorns vary discontinuously in stamen number per flower in a manner correlated with ploidy level such that the derived number (5–10) has only been found in polyploids, while diploids have been shown so far to have exclusively 15–20 (or more) stamens per flower (Table 2). With a single exception known to date, Eurasian *Crataegus* species have 15–20 stamens per flower regardless of ploidy level (*Christensen, 1992*).

## DNA extraction, sample preparation, and sequencing

DNA was extracted from dried tissue using the FastDNA Spin Kit from MPBiomedicals (Santa Ana, CA, USA) following manufacturer's protocols, modified by the addition of 40 μL 1% polyvinylpyrrolidone and 10 μL β-mercaptoethanol to the lysis solution prior to grinding. Aliquots of isolated DNA were sheared with a BioRuptor® Pico sonicator (Diagenode Inc., Denville, NJ, USA) for 30 cycles of 30 s on, 30 s off. Libraries for sequencing were prepared using the NEBNext® Ultra™ DNA Library Prep Kit for Illumina® (New England Biolabs, Ipswich, MA, USA). Samples were dual indexed using both the i5 and i7 primers from the NEBNext® Multiplex Oligos for Illumina® (Dual Index Primers Set 1).

A set of probes targeting 257 conserved, putatively orthologous nuclear loci (Table S1) were designed from the apple, peach and strawberry genomes (*Liston, 2014*). The probes designed from the *Fragaria vesca* genome (*Shulaev et al., 2010*) have been previously used to conduct phylogenetic analysis and resolve the origin of polyploid species in *Fragaria* (*Kamneva et al., 2017*), *Lachemilla* (*Morales-Briones, Liston & Tank, 2018*), and *Rubus* (*Carter et al., 2019*). For this study, we used the probes from the apple (*Malus × domestica* 'Golden Delicious') genome (*Velasco et al., 2010*). Samples were pooled by equal molarity and enriched for the targeted loci following the Mybaits version 2.3.1 protocol (MYcroarray, Ann Arbor, MI, USA). Enriched products were sequenced using 75 bp paired-end reads on an Illumina® MiSeq (Illumina Inc., San Diego, CA, USA) at the Center for Genome Research and Biocomputing at Oregon State University.

## Bioinformatics

Nucleotide sequences were quality trimmed (Q15 on left, Q10 on right) and Illumina adapters were removed using cutadapt (*Martin, 2011*). Duplicated reads were reduced by calculating coverage of unique reads in assembly and reducing duplicated reads to represent the same coverage. Cleaned reads were assembled with HybPiper (*Johnson et*

**Table 2  Intermediacy of allotetraploid *Crataegus douglasii* Lindl. and *C. rivularis* Nutt. with respect to their diploid progenitors in *C.* subg. *Sanguineae* Ufimov and sympatric tetraploid members of *C.* subg. *Americanae* El-Gazzar.** Ploidy level data as in *Talent & Dickinson (2005)* and *Zarrei et al., (2015)*. Morphological and ecological data summarized from T. A. Dickinson, unpubl. mss; descriptors as in *Dickinson et al. (2008)*. Geographic distribution (Canada, United States) as per *Phipps (2015)*.

**Allotetraploid *Crataegus douglasii***

|  | C. subg. *Sanguineae*<br>*C. suksdorfii* (2*x*) | allotetraploid | C. subg. *Americanae* (4*x*)<br>(*C. chrysocarpa* s.l.,<br>*C. macracantha* s.l.) |
|---|---|---|---|
| Thorn length | Short | Intermediate | Long |
| Leaf toothing | 5—7/1.0 cm | 8—10/1.0 cm | 7—12/1.0 cm |
| Calyx lobe length | Short | Intermediate | Long |
| Calyx lobe margination | Teeth absent | Intermediate | Teeth abundant |
| Stamens per flower | Ca. 20 | Ca. 10 | Ca. 10 |
| Styles per flower | (3-) 4—5 (-6) | 3—4 (-5) | 2—3 (-5) |
| Ecological amplitude | Narrow | Broad | Broad |
| Geographic range | Northern CA, western OR, southwestern WA | AB, BC, CA, ID, MT, ON, OR, WA | Nearly transcontinental |

**Allotetraploid *Crataegus rivularis***

|  | C. subg. *Sanguineae*<br>*C. saligna* (2x) | allotetraploid | C. subg. *Americanae* (4*x*)<br>(*C. chrysocarpa* s.l.,<br>*C. macracantha* s.l.) |
|---|---|---|---|
| Thorn length | Short | Short | Long |
| Calyx lobe pubescence | Absent | Sparse | Abundant |
| Calyx lobe margination | Teeth few or absent | Intermediate | Teeth abundant |
| Stamens per flower | Ca. 20 | Ca. 10 | Ca. 10 |
| Styles per flower | 4—5 | 3—4 (-5) | 2—4 (-5) |
| Ecological amplitude | Narrow | Broad | Broad |
| Geographic range | Western CO, northeastern UT | AZ, CO, ID, NM, NV, UT, WY | Nearly transcontinental |

*al., 2016*) using default parameters. The coding regions (CDS) of the 257 apple genes (*Velasco et al., 2010*) were used as read mapping targets, with ambiguous bases in the apple sequence (0.2% the total bases) replaced with Ns. The CDS sequences were assembled for all individuals at each locus. Multiple sequence files for all individuals and diploids only were aligned with MAFFT v7.402 (*Katoh & Standley 2013*). We used the MAFFT default alignment settings and the "auto" option which selects an appropriate algorithm according to data size (Tables S2, S3). All alignments were visually inspected, and those with >20 bp of non-homologous sequence in two or more samples were flagged as poor-quality alignments. If misaligned regions were found at the end or ends of the alignments, they were deleted (but only if that left 600 bp or more). In addition, alignments with two or more sequences identified by HybPiper as potential paralogs were noted.

## Phylogeny inference from nuclear data

Phylogenetic relationships were estimated with IQ-TREE v. 1.7-beta7 (*Nguyen et al., 2015*). First a concatenation of all included nuclear gene alignments was used to reconstruct a maximum-likelihood (ML) species tree using an edge-linked proportional partition model

and 1,000 rapid bootstrap replicates. Next, an ML gene tree was reconstructed for each locus. Best fit substitution models were obtained with the ModelFinder option (*Kalyaanamoorthy et al., 2017*), while branch support was obtained using the ultrafast bootstrap (*Hoang et al., 2018*) and the SH-aLR test (*Guindon et al., 2010*; *Minh et al., 2020b*). To protect against overestimating branch support because of severe model violations we used the option to optimize the bootstrap trees by nearest-neighbor interchanges in the bootstrap alignments (*Minh et al., 2020b*). Finally, the concordance and discordance factors were calculated by IQ-TREE for each branch in the 14- and 24-accession trees (*Lanfear, 2018*; *Minh, Hahn & Lanfear, 2020a*). The gene concordance factor (gCF) is defined as the percentage of decisive gene trees containing that branch. The site concordance factor (sCF) is defined as the percentage of decisive sites supporting a branch. Discordance factors quantify the amount of disagreement among loci and sites, and are defined as the percentages of genes (gDF) and sites (sDF) supporting alternative resolutions (nearest-neighbor interchanges) of a given branch (*Minh, Hahn & Lanfear, 2020a*). The gene discordance factors are the percentages of decisive trees supporting a second, alternative resolution of the four clades around this branch ($gDF_1$), and the percentages of decisive trees supporting a third, alternative resolution of the four clades around this branch ($gDF_2$). Analogously, the site discordance factors answer the question, ''which of the three possible quartets around a given branch'' does a site support (*Lanfear, 2018*)? These are calculated using the number of decisive sites (sN) averaged over many possible quartets partitioned between the one supporting the tree obtained (sCF) and those supporting the next two best resolutions ($sDF_1$, $sDF_2$). We also carried out a test of $H_0$: incomplete lineage sorting (ILS) is responsible for approximately equal numbers of genes supporting alternative topologies (*Lanfear, 2018*) using the data and graphics analysis environment R (*R Core Team 2016*). In fact, except as noted, R *packages* and **functions** were used for all of the data manipulations and analyses described below.

The multi-species coalescent model of ASTRAL-III v. 5.6.3 (*Zhang et al., 2018*) was also used to estimate phylogenetic relationships for the diploid species and diploid plus polyploid species using 244 nuclear loci found in all samples, and 245 nuclear loci found in all diploid samples. Sequence obtained from the *Malus × domestica* 'Golden Delicious' genome (*Velasco et al., 2010*) was specified as the outgroup for the above analyses.

### Phylogeny inference from plastome data

Plastome sequences were obtained from the unenriched fraction of target capture libraries (*Weitemier et al., 2014*). To assemble the plastomes, SPAdes v. 3.6.0 (*Bankevich et al., 2012*) was used for de novo assembly of the cleaned reads from each sample. Resulting scaffolds were aligned to the *Malus × domestica* 'Golden Delicious' plastome sequence (*Velasco et al., 2010*) with BLAT (*Kent, 2002*). One copy of the inverted repeat was removed from the apple sequence before alignment. The multiple alignment files were imported into Geneious v. 6.1.8 (*Kearse et al., 2012*) and manually refined. Consensus sequences were output and aligned with MAFFT v7.312. Gblocks v 0.91b (*Talavera & Castresana, 2007*) was used to remove ambiguously aligned sequence (block size of 10, maximum of 8 non-conserved positions or >50% gaps). Maximum likelihood estimation for the plastome phylogeny

was conducted with IQ-TREE (*Nguyen et al., 2015*) as described above for the individual nuclear loci. The plastome was treated as a single locus for phylogenetic analysis, in accord with *Doyle (2021)*.

## Linkage group localization

We used the BLAST functionality of the Genome Database for Rosaceae (*Jung et al., 2019*) to localize to their apple chromosome the apple loci whose *Crataegus* counterparts we recovered, using both the GDDH13 v1.1 (*Daccord et al., 2017*) and the HFTH1 Genome v1.0.a1 (*Zhang et al., 2019*) chromosome databases.

## Phylogenetic invariants

We also used our concatenated nuclear sequence alignment (529,827 sites) to test hypotheses of *C.* subg. *Sanguineae* × *C.* subg. *Americanae* hybridization for *C. douglasii* and *C. rivularis* using coalescent-based phylogenetic invariants and the software package HyDe (*Blischak et al., 2018*). In this way we tested $H_0$: the admixture statistic ($\gamma$) = 0 (*i.e.,* no admixture). After converting the sequence data from FASTA to PHYLIP format with Fasta2Phylip.pl (*Deng, 2007*) or the **dat2phylip** function in the *phylotools* package (*Zhang, 2017*), for each species a selection of 13 accessions (four outgroups, three supposed hybrids, two diploid subg. *Sanguineae* parents, and four subg. *Americanae* tetraploid parents; Table 1) was tested at the level of populations and individuals, the latter with and without bootstrapping. Distributions of the admixture statistic ($\gamma$) were plotted as suggested by *Kabacoff (2017)*. To provide context for these results we also used HyDe to test for any signal of hybridization in the sequence data for tetraploid *C. macracantha* using the single accession of diploid *C. calpodendron* as one parent (P1), and either the remaining four *Americanae* diploids or the four *Sanguineae* diploids to represent the other parent (P2; Table 1). *Crataegus calpodendron* and *C. macracantha* are both placed in *C.* sect. *Macracanthae*, the smaller of the two sections in subg. *Americanae* that is distinguished in part by the excavations on the radial surfaces of the pyrenes, much as are taxa in subg. *Sanguineae* (Table 1; *Phipps, 2015*). The four remaining *Americanae* diploids all belong to different series in the large section *Coccineae* (Table 1; *Phipps, 2015*). A similar examination of tetraploid *C. chrysocarpa* was not carried out because there were no other species in our sample from *C.* ser. *Rotundifolieae*, let alone diploid ones, to serve as at least one plausible parent.

## Splits network

The single locus sequence alignments in FASTA format of 244 low copy number nuclear loci (529,827 positions) available for the diploid plus polyploid species sample were concatenated into a single multilocus FASTA alignment with the perl script catfasta2phyml (*Nylander 2020*). This multilocus alignment was used to produce a Splits network representation of the 24 hawthorn individuals (Table 1) using SplitsTree4 (*Huson & Bryant, 2005*) and the default NeighborNet parameters. The diagram was calculated from the uncorrected p-distances between the aligned sequences, with a fit = 97%. Sequences from the apple (*Malus × domestica* Borkh.) genome (*Velasco et al., 2010*) were included.

The diagram was colored using ColorBrewer 2.0 (*Brewer 2013*) and Adobe Illustrator 2020 (*Adobe Inc. 2019*).

## Gene tree comparisons

We evaluated the congruence of the gene trees we obtained, the extent to which they suggest that inferred *C.* subg. *Sanguineae* allotetraploids contain alleles indicative of their parentage, and the support these trees provide for the a priori classification of the sample based on earlier work (Fig. 2; Table 1). Using just the trees for our diploids-only sample, we tabulated taxonomically significant features of the reference trees (plastome and nuclear multilocus trees) and the single-locus trees, namely (a) sister-group relationships of the early-diverging subgenera, (b) the extent to which subgenera *Americanae* and *Sanguineae* formed separate clades, and (c) the extent to which accessions belonging to the taxonomic sections within the focal group for our study, *C.* subg. *Sanguineae*, also formed distinct clades. We then examined how the topologies of the single-locus diploid-only trees best representing the infrageneric classification were impacted, in parallel analyses, by the inclusion of allotetraploid accessions. Except as noted, all analyses and manipulations of our trees were carried out in R (*R Core Team, 2016*) using the R package *ape* (*Paradis et al., 2015*).

The taxonomic structure in our sample provides two sets of categories to which our accessions belong, namely the subgenera and sections of the genus *Crataegus* (Table 1). Thus we also used the **treeConcordance** function in the R package *treespace* (*Jombart et al., 2017*) to measure the extent of the agreement between each of our single-locus trees and the hierarchical classification implied by the two sets of categories as seen in our plastome trees (calculated using the **makeCollapsedTree** function; Fig. 2). This was carried out separately for the sample of diploid accessions only, and for the diploid plus the allotetraploid accessions (Table 1). Ordering the trees according to their tree concordance values allowed us to compare highly concordant trees with ones that were only minimally so. The former were defined as exceeding the third quartile by more than 1.5× the interquartile range of the concordance values (documentation for the **boxplot.stats** function; *R Core Team, 2016*), whereas the latter comprised the first quartile of concordance values. In order to distinguish the Kendall et al. tree concordance values from the gCF and sCF concordances calculated using IQ-TREE (above), we abbreviate these concordances as rtCF, referring to the related sets of tips (categories) that make the measure possible.

We used the *treespace* function **relatedTreeDist** to calculate the Related Tree (RT) distances between trees (*Kendall, 2019*; *Kendall, Eldholm & Colijn, 2018*). This calculation involves comparing the collapsed forms of two trees with respect to whether the categories (here, subgenera or sections) in one or both remain monophyletic, so that the distance quantifies similarities and differences between the trees relative to phylogenetic relationships between the categories (*Kendall, 2019*).

In addition, we used three descriptors of tree topology (Cluster Membership Divergence, CMD; Subtree Membership Divergence, SMD; and Partition Membership Divergence, PMD), singly and together (*Podani, 2000*; *Podani & Dickinson, 1984*). Trees described in the Newick format (*Felsenstein, 2005*) were converted to merge matrices (*Hartigan,*
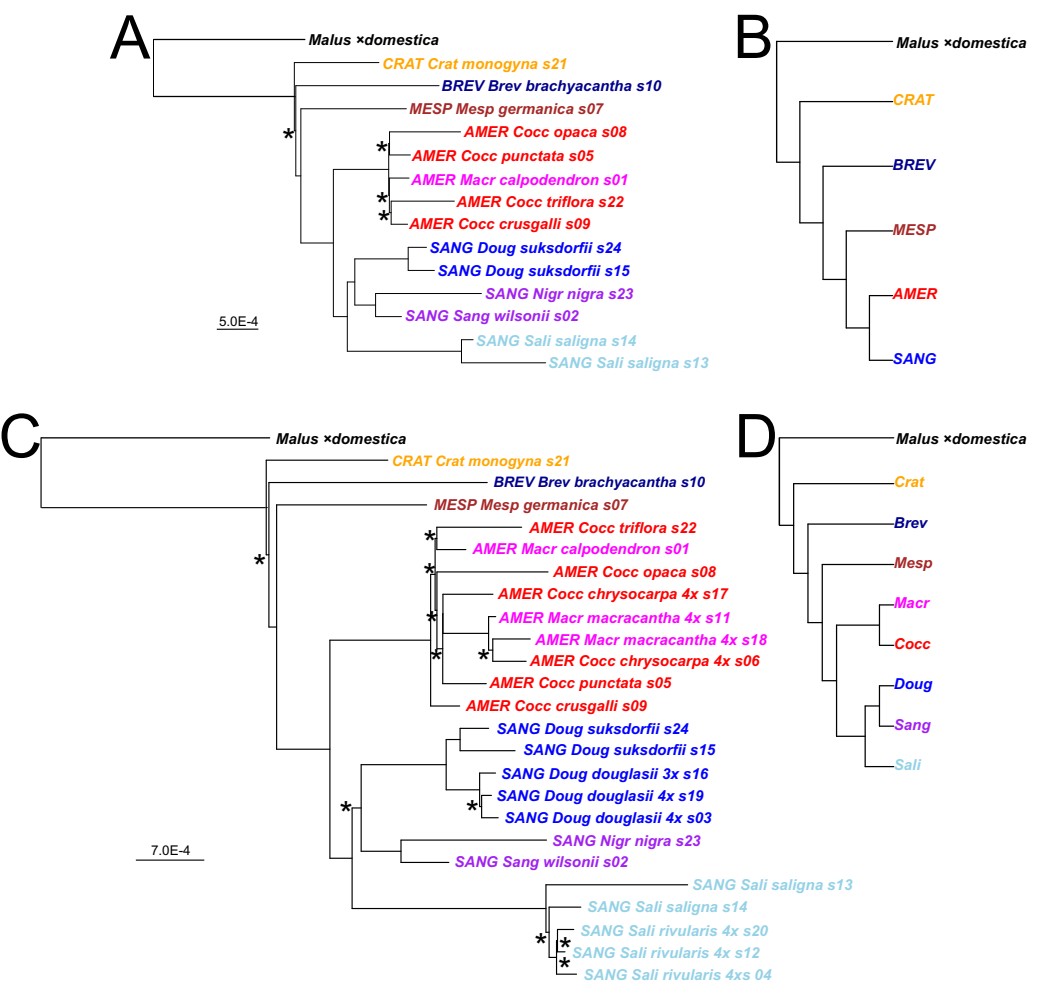

**Figure 2** *Crataegus* **infrageneric classification and impact on tree topology of including allotetraploids in a plastome phylogeny.** (A) The RAxML plastome tree for 14 diploid *Crataegus* accessions (numbered as in Table 1) coded to show subgenera (*MESP*, *BREV*, *CRAT*, *AMER*, SANG) and sections (color): *Crataegus* (*Crat*), *Brevispinae* (*Brev*), *Mespilus* (*Mesp*), *Coccineae* (*Cocc*), *Macracanthae* (*Macr*), *Salignae* (*Sali*), *Douglasianae* (*Doug*), and *Sanguineae* (series ***Sang**uineae* and ***Nigr**ae*). (B) The plastome tree in (A) collapsed so as to show the topological relationships between the five subgenera. (C) The RAxML plastome tree for the same 14 diploid and 10 related tetraploid *Crataegus* accessions (Table 1). (D) The plastome tree in (C) collapsed so as to show the eight taxonomic sections of *Crataegus* represented in our sample. In (B) and (D) subgenera are color-coded as in Fig. 1. All trees have been rooted using the plastome of apple, *Malus ×domestica* (*Velasco et al., 2010*). In (B) and (D), collapsing was done using the function **makeCollapsedTree** in the R package *treespace* (*Jombart et al., 2017*). Nodes have bootstrap support ≥ 96% (diploids only) or ≥ 95% (diploids + tetraploids) unless indicated otherwise (*); scale bars for branch lengths are in substitution units.

*1967*; *Podani, 1982*) using program NTOS.EXE in the SYN-TAX 5.10 package (*Podani, 1993*), having first replaced taxon labels with numbers, and branch lengths with arbitrary values. Merge matrices were concatenated and used as input to program DENDAT.EXE, in the SYN-TAX package, in order to calculate CMD, PMD, and SMD. Here, the number of OTUs in the tree is $N = 15$ or $N = 25$, so the number of pairwise comparisons is

$w = N \times (N-1)/2$, so that $w = 105$ or $300$ for each descriptor. We are comparing $k$ trees (including the plastome tree), so for each descriptor we obtain a data matrix with $w$ rows and $k$ columns and, for the three descriptors taken together, a matrix of $3w$ rows and $k$ columns. These matrices were used as input to calculate Euclidean multivariate (MV) distances between all pairs of the $k$ trees using the R function **dist**. For more details, see *Podani & Dickinson (1984)* who explained the methodology with reference to a set of artificial trees, a set of dendrograms from a phenetic study of *Crataegus*, and a set of cladograms from a published molecular phylogeny of mammals.

We calculated distances between our trees in order to display the variation they exhibit in the low-dimensional spaces of the first Principal Coordinates Analysis (PCoA) axes for which the proportion of variance accounted for by the axis exceeds that expected under the broken-stick model (*Frontier, 1976*; *Legendre & Legendre, 1998*). Displayed in this way, we superimposed information about the rtCF values and distinguished between ones found to be highly concordant and others with the lowest rtCF values. PCoAs were calculated using the function **pcoa** in the R package *ape* (*Paradis et al., 2015*). We also clustered the trees based on the RT and MV distances between them by using the R function **hclust**. Scree plots of the PCoA eigenvalues were made from the output of the **pcoa** function.

## RESULTS

### Target enrichment and plastome assembly

An average of 3.06 million reads were obtained per sample with an average 25.6% of reads on targeted nuclear loci. On average, 254.6 genes were recovered per sample, and 244 assemblies had sequence data for all samples, both diploid and tetraploid (Table 1). Assembled loci average 1,262 bp and cover 81% of the target sequence. No ambiguous bases are present in the assembled sequences. The average total sequence length is 424.5 kbp per sample. The median size of assembled plastomes was 130.7 kbp. Four samples with plastome read coverage below 4.5× had incomplete assemblies, ranging from 112.6 kbp to 123.0 kbp. The remaining samples with coverage above 4.5× had nearly complete assemblies.

### Linkage group localization

The 257 loci recovered represent all 17 apple chromosomes, with six or more loci found on each chromosome, and 20 or more loci on each of chromosomes 6, 11, 15, and 17 (Table S1). Correspondingly fewer than the equidistribution localized to the remaining chromosomes. Six loci localized to unanchored scaffolds ("Chr00" Table S1) in the Golden Delicious genome. Locus 99 found no hits on any of the scaffolds (Table S1).

### Plastome trees

Plastome trees were generally well-supported in terms of bootstrap values as they relate to the subgenera and to the better-sampled sections of *C.* subg. *Sanguineae*, regardless whether they were obtained from the diploids-only sample (Fig. 2A), or the diploids+polyploids sample (Fig. 2C). In the plastome trees, regardless of the sample, support for the branching order of subgenera *Crataegus* and *Brevispinae* was weak (Figs. 2A, 2C). Support, however,

was strong for the sister-group relationship between subg. *Mespilus* and the rest of the genus (100%). Monophyly of each of subgenera *Americanae* and *Sanguineae* was well-supported (100%). These latter subgenera differ, however, in how much the sections they comprise are distinct from each other. Sections *Douglasianae*, *Salignae*, and *Sanguineae* are represented by two diploid accessions each (Table 1), and each pair has 100% bootstrap support for both the diploids-only and the diploids+polyploids samples (Figs. 2A, 2C). Less importance was attached to the sampling of subg. *Americanae*, so that four of the five diploids from this subgenus each belonged to different series within *C.* sect. *Coccineae* (Fig. 2A; Table 1; the fifth diploid belongs to sect. *Macracanthae*). As a result, in the diploids-only sample there was no opportunity to form clades representing taxonomic groups, and in fact branch support for individual *Americanae* accessions was ≤ 67% (Fig. 2A). Similarly, in the subg. *Americanae* diploids+polyploids sample there was no well-supported taxonomic structure, unlike that seen in the corresponding subg. *Sanguineae* sample (Fig. 2C).

## Gene trees

Sequence assembly yielded alignments for the entire plastome, less one copy of the internal repeat, and for 257 low copy-number nuclear loci. The latter comprised 25 characterized as poor-quality alignments and were edited as described in the methods. Only six loci were flagged by HybPiper as containing paralogs, and these include two that were characterized as poor-quality alignments.

Twelve loci were eliminated because they lacked data for one or more terminals in the diploids-only sample. A thirteenth was eliminated from the diploids+polyploids sample for the same reason. For the sample of 14 diploid *Crataegus* accessions only, gene trees were considered only for the 245 nuclear loci for which all 14 accessions were represented (Fig. S1; Table S2). For the sample of 24 diploid and tetraploid accessions, only 244 gene trees were considered (Fig. S1; Table S3), for the same reason. Comparison was also made with the multilocus coalescent tree for all 244 or 245 loci (Fig. 3).

The first criterion used in tabulating and sorting the trees for the diploid accessions concerned the extent to which trees conformed to the subgeneric topology seen in earlier work (*Lo & Donoghue, 2012*; *Lo et al., 2009a*; *Lo, Stefanović & Dickinson, 2007*; *Zarrei et al., 2015*), and in the plastome tree (Fig. 2). In the earlier work *C. germanica* (western Eurasia) and *C. brachyacantha* (North America) were found to be sister to the remainder of the genus and, within that remainder, *C.* subg. *Crataegus* (western Eurasia) was sister to the clade comprising *C.* subg. *Americanae* (North America) and *C.* subg. *Sanguineae* (eastern Eurasia and North America). Here, however, in the plastome tree (Fig. 2; Fig. 1 in *Ufimov & Dickinson, 2020*) and in trees for 12 of the nuclear loci *C. monogyna* is sister to the remainder of the genus (*e.g.*, Dtree17, Dtree58; Fig. S1). In the trees for 35 loci *C. brachyacantha* was sister to the remainder of the genus (*e.g.*, Dtree208; Fig. S1). *Crataegus germanica* was sister to the remainder of the genus in 97 single locus trees (all remaining diploids-only trees in Fig. S1), and in the multilocus trees calculated over all 244 or 245 loci (Fig. 3). In the trees for 16 loci a clade comprising *C. germanica* and *C. brachyacantha* was sister to the rest of the genus (Table S2). In 46 trees, various *Americanae* and *Sanguineae* accessions formed two- to four-member clades with each other, or with one or more of

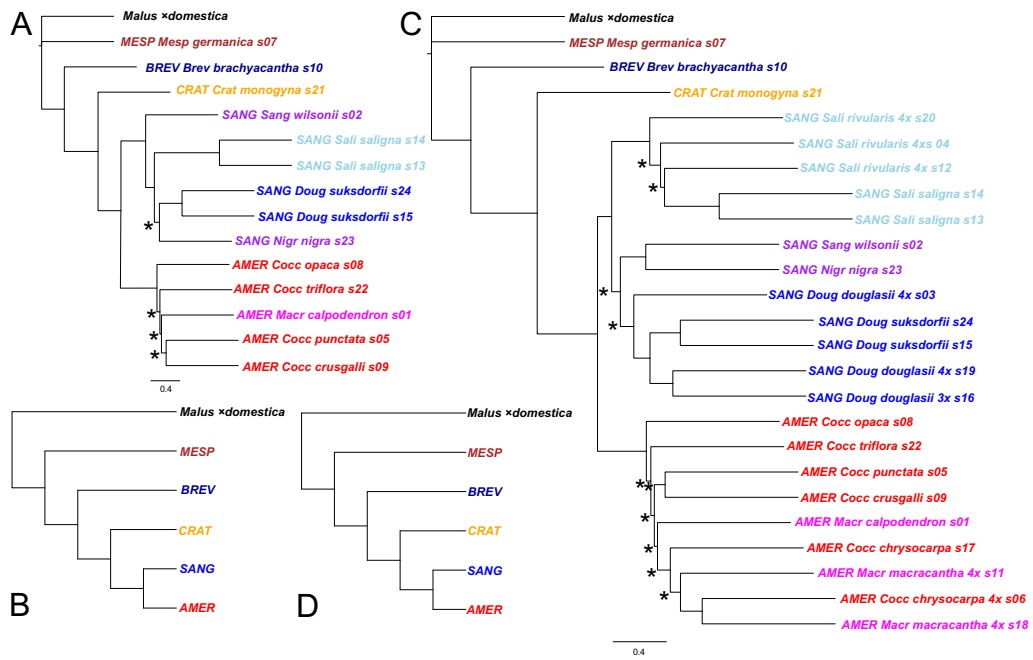

**Figure 3** **Impact on tree topology of including allotetraploids in a multilocus coalescent phylogeny of diploid *Crataegus* accessions.** (A) ASTRAL-III tree for 14 diploid *Crataegus* accessions (numbered as in Table 1) based on sequence data for 245 low copy number nuclear loci; (B) the tree in (A) collapsed by subgenera using the function **makeCollapsedTree** in the R package *treespace* (*Jombart et al., 2017*); (C) ASTRAL-III tree for the same 14 diploids plus 10 related allotetraploid *Crataegus* accessions (Table 1) based on sequence data for 244 low copy number nuclear loci; (D) the tree in (C) collapsed by subgenera using the R function **makeCollapsedTree**. In (A) and (C) trees rooted using the corresponding sequences from the genome of apple, *Malus ×domestica* (*Velasco et al., 2010*). Accessions are coded as in earlier figures by *Crataegus* subgenera (*MESP, BREV, CRAT, AMER, SANG*) and sections (color; *Mesp, Brev*, Crat, *Cocc, Macr, Doug, Sali*, and series **Sang**uineae and **Nigr**ae in section *Sanguineae*; Table 1). In (A) and (C) nodes have local posterior probability support ≥ 0.95 unless indicated otherwise (*); scale bar for branch lengths in coalescent units (*Sayyari & Mirarab, 2016*).

*C. brachyacantha*, *C. germanica*, or *C. monogyna* and these were sister to the rest of the genus (Table S2). Finally, in the 39 remaining trees the topology or the resolution made meaningful assessment of sister group relationships impractical (Table S2). The remaining criteria, concerned with manifestation of the taxonomic structure of our sample, as seen in the topology of the single-locus trees, are discussed below.

Only 19 of the single-locus diploids-only trees showed markedly high rtCF values, exceeding the third quartile by more than 1.5 × the interquartile range of these values (concordance with the plastome tree 0.573–0.750; Fig. S1). These trees formed the basis for further analyses of the impact seen on the topology of the diploid-only trees from adding in the tetraploid accessions (Fig. 4; Fig. S1), and were derived from 16 "good" and three "medium" quality alignments (Table S1).

Locus 95 (MDP0000220167, on chromosome 8, probable Vacuolar proton translocating ATPase 100 kDa subunit; Figs. S1, S2; Table S1) was the only one of these 19 loci to produce a tree for the diploid+tetraploid sample in which *C.* subg. *Americanae* and subg. *Sanguineae*

 

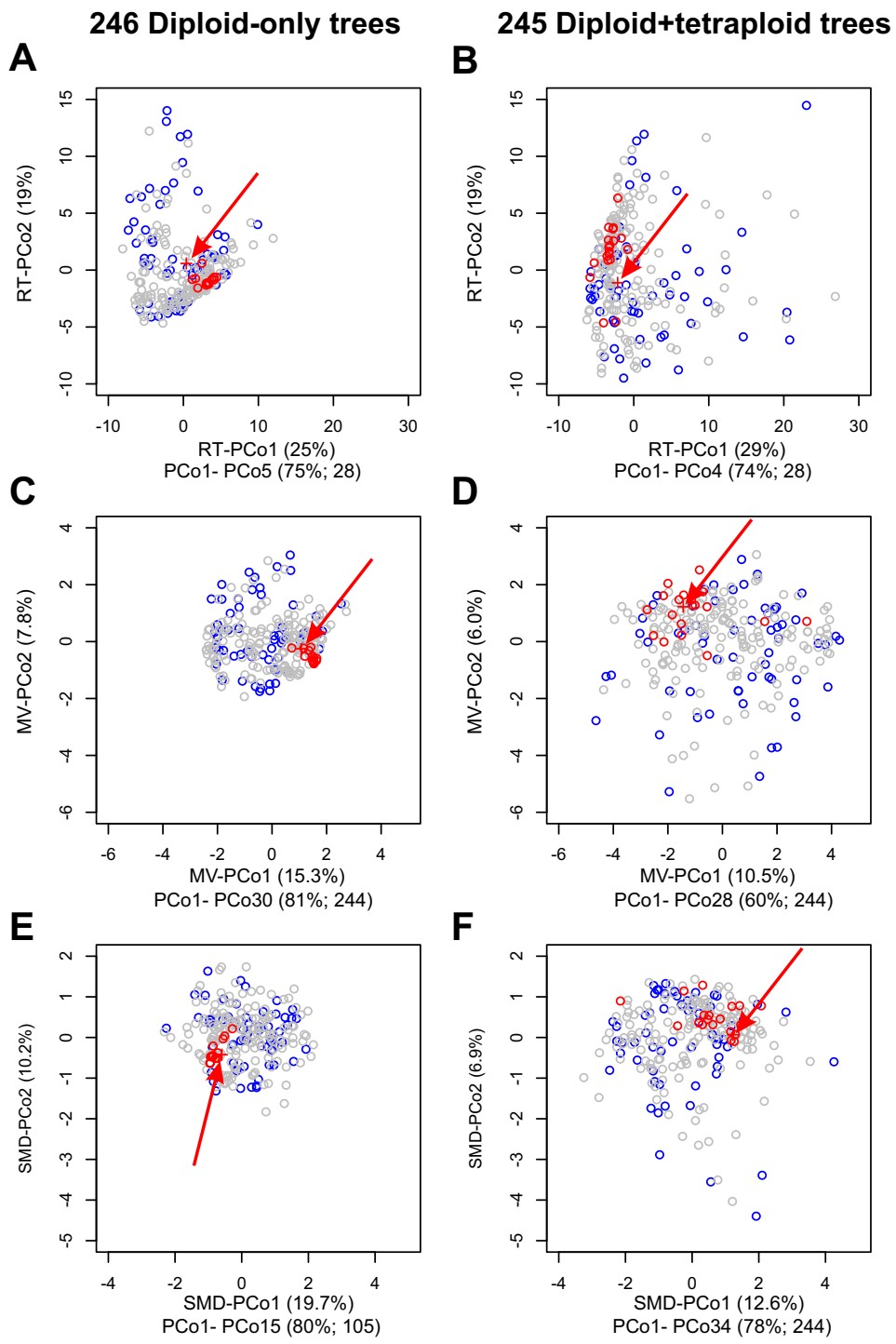

**246 Diploid-only trees**  **245 Diploid+tetraploid trees**

**Figure 4** **Impact on tree topologies of including allotetraploids on Principal Coordinates Analyses (PCoAs) of distances between plastome and 245 nuclear single-locus trees for diploid *Crataegus* accessions.** (A, C, E) PCoA of 246 trees calculated for the diploid accessions only (plastome tree + 245 gene trees). 

**Figure 4 (…continued)**
(B, D, F) PCoA of 245 trees calculated for the diploid and tetraploid accessions (plastome tree + 244 gene trees). (A, B) PCoA calculated from a matrix of related tree (RT) distances between the trees. (C, D) PCoA calculated from a matrix of multivariate (MV) distances between the trees. (E, F) PCoA calculated from a matrix of subtree membership divergence (SMD) distances between the trees. In each figure the position of the plastome tree is indicated by a plus sign (arrowed). Symbol color reflects the concordance between a gene tree and the corresponding plastome tree, collapsed with respect to the sectional affiliation of the accessions: red, trees within the fourth quartile of the concordances that are outliers; blue, trees in the first quartile of the concordances; all other trees gray. Note that the outgroup, apple, was used to root all trees in order to ensure topological consistency and facilitate visual comparison (Fig. S1).The relative magnitudes of the eigenvalues in (A–F) are depicted in scree plots (Fig. S4).

were reciprocally monophyletic, demonstrating that for this alignment inclusion of the tetraploid accessions in estimating the gene phylogeny had little impact on either the topological relationships between the subgenera or on the memberships of these two clades (Fig. S2 ; Table S3). Support is high for the nodes supporting the early-diverging subgenera (*Brevispinae*, *Crataegus*). However, support is low for nodes supporting the *Americanae* and *Sanguineae* clades in both the diploids-only and the diploids+tetraploids trees (Figs. S1, S2; Tables S2, S3). Although the three taxonomic sections belonging to *C.* subg. *Sanguineae* also formed distinct subclades (compare Fig. 2 and Tree5 in Fig. S1; Tables S2, S3), support was nevertheless low for most of the internal nodes in these clades. In the remainder of the 19 trees the principal impact on tree structure of including the inferred *Sanguineae* allotetraploids and the widespread *Americanae* tetraploids (*C. chrysocarpa*, *C. macracantha*) was that in 16 of them one to five of the *Sanguineae* tetraploids were included in the *Americanae* clade (Fig. S1). In the trees for loci 222 and 254 one or two of the *Americanae* tetraploids was included in the *Sanguineae* clade. In several instances some *Sanguineae* diploids were also included in the *Americanae* clade (*e.g.*, Tree179, Tree198; Fig. S1).

### Phylogenetic invariants
Calculating the amount of admixture ($\hat{\gamma}$; Table 3; Fig. S3) from our nuclear sequence data strongly suggests that *C. douglasii* and *C. rivularis* are intersubgeneric hybrids ($\hat{\gamma}$ values in the ranges 0.6–0.76 and 0.69–0.77, respectively, associated with vanishingly small *p*-values; Table 3). In contrast, for the putative hybrid tetraploid *C. macracantha*, support for $H_0$: $\gamma = 0$ was indicated by low or modest values of $\hat{\gamma}$ (P2 = *Americanae*, 0.06; P2 = *Sanguineae*, 0.31), associated with correspondingly high *p*-values (0.4 and 1.0, respectively).

### Splits network
All of the *Sanguineae* tetraploid samples are represented by reticulate lines, consistent with their hybrid origin. The absence of reticulation in subgenus *Americanae* is an artifact of our limited sampling of this large subgenus. Diploids in subgenus *Sanguineae* also show reticulation presumably due to their shared ancestry with the tetraploids.

### Multivariate comparisons of gene trees
Principal Coordinates analyses (PCoA) of our samples of diploids-only trees, as represented by the MV and RT distances between them (Fig. 4), demonstrate (a) clustering of the trees

**Table 3 Results of population- and individual-level hybridization detection analyses using HyDe (Blischak et al. 2018).** For each putative allotetraploid (Hybrid), Parent 1 and Parent2 are *Crataegus* subg. *Sanguineae* diploids[1,3] and *C.* subg. *Americanae* tetraploids[2], respectively (Table 1). Outgroups were the apple, *C. monogyna* (s21), *C. germanica* (s07), and *C. brachyacantha* (s10) accessions (Table 1). Results are shown for *C. douglasii* and *C. rivularis* at the population level, and then the individual level (Table 1), where $\hat{\gamma}$ is the admixture statistic, tested for $H_0$: $\gamma = 0$ versus $H_1$: $\gamma > 0$.

| Parent1 | Hybrid | Parent2 | Z-score | *P*-value | $\hat{\gamma}$ |
|---------|--------|---------|---------|-----------|------|
| SANG[1] | *C. douglasii* | AMER[2] | 7.287 | $1.594 \times 10^{-13}$ | 0.67244 |
| SANG[1] | s03dou | AMER[2] | 7.815 | $2.776 \times 10^{-15}$ | 0.60647 |
| SANG[1] | s16dou | AMER[2] | 6.017 | $8.938 \times 10^{-10}$ | 0.75688 |
| SANG[1] | s19dou | AMER[2] | 7.676 | $8.216 \times 10^{-15}$ | 0.64428 |
| SANG[3] | *C. rivularis* | AMER[2] | 10.001 | 0.0 | 0.73466 |
| SANG[3] | s20riv | AMER[2] | 10.011 | 0.0 | 0.69301 |
| SANG[3] | s04riv | AMER[2] | 9.751 | 0.0 | 0.77420 |
| SANG[3] | s12riv | AMER[2] | 10.178 | 0.0 | 0.72819 |

**Notes.**
[1] *C. suksdorfii* s15, s24.
[2] *C. chrysocarpa* s06, s17, *C. macracantha* s11, s18.
[3] *C. saligna* s13, s14.

most closely concordant with the plastome tree; (b) dispersion along the PCoA axes interpretable with respect to features of tree topology and clade composition (Figs. S1, S2 and Tables S2, S3); and (c) the way in which tree concordance and the distances between trees may reflect different features of the trees (*Kendall, Eldholm & Colijn, 2018*). It is also apparent from scree plots (Fig. S4) of the eigenvalues represented by the PCoA axes in these ordinations that the RT distances summarize the variation among our diploids-only trees in fewer dimensions (five significant axes, accounting for 75% of the total variance) than was the case with the MV distances (30 significant axes, accounting for 81% of the total variance; the first five axes account for only 39%).

## Multilocus trees

Both the coalescent and concatenated multilocus trees were all highly concordant with the plastome trees, whether for the diploids-only or the diploids+teraploids sample (rtCF values in the ranges 0.62–0.78, in comparisons with the corresponding subgenus- and section-collapsed plastome trees; Figs. 2B, 2D). Well-supported differences between the ASTRAL-III trees (Fig. 3) and the plastome trees (Fig. 2) were restricted to branching order relationships between the early-diverging subgenera *Mespilus*, *Brevispinae*, and *Crataegus* (compare Figs. 2 and 3). The sister-group relationship between subgenera *Americanae* and *Sanguineae* was strongly supported, as was the monophyly of each of the three sections in subgenus *Sanguineae*. Sections *Coccineae* and *Macracanthae* in subgenus *Americanae* were not monophyletic with our sample (Fig. 3).

IQ-TREE provides not only bootstrap support values, but also gene (gCF) and site (sCF) concordances and discordances (gDF and sDF, Fig. 5; *Lanfear, 2018*; *Minh, Hahn & Lanfear, 2020a*; *Minh et al., 2020b*). These values (for the 12 nodes shared by the diploid-only and the diploid + tetraploid trees; compare Figs. 5A–5C) demonstrate how inclusion of the tetraploid hybrids and their *Americanae* parents in trees reduces the proportion of gene trees supporting individual nodes (gCF) from 5–65% in the diploids-only sample to 1–40%

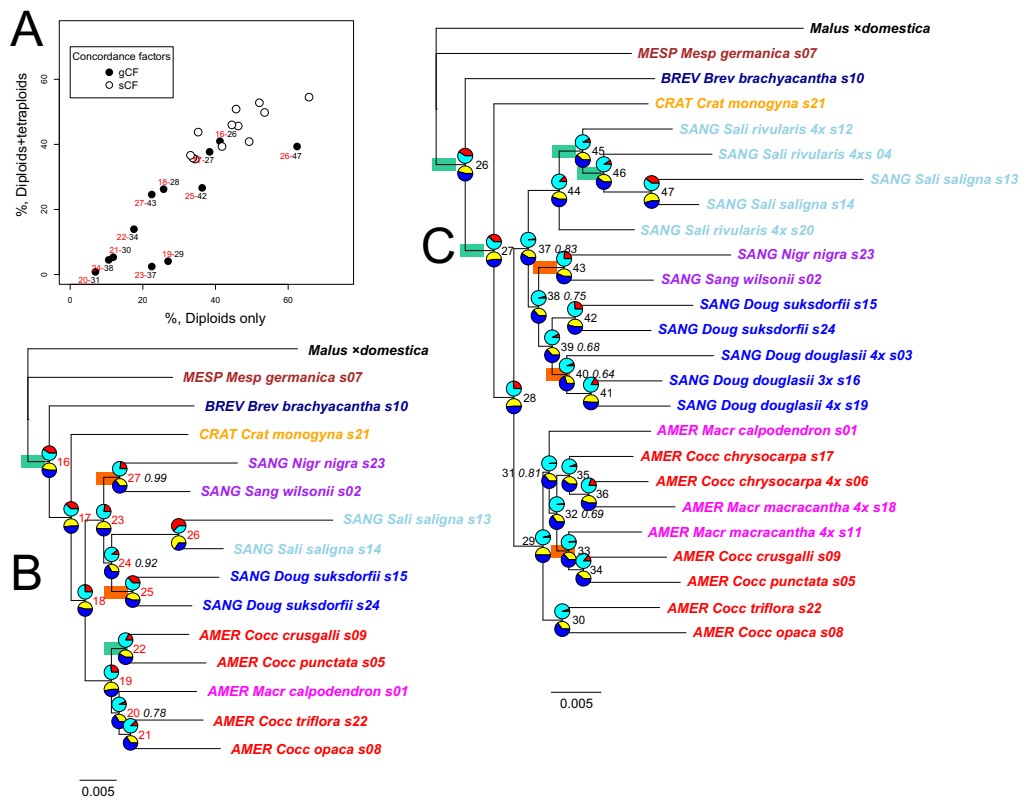

**Figure 5** **Impact on gene and site concordances of including allotetraploids in the multilocus ML phylogeny of diploid _Crataegus_ accessions.** (A) Comparison of gene tree (gCF) and site (sCF) concordance values (Minh et al. 2020; (_Nguyen et al., 2015_)) for nodes shared between the IQ-TREE concatenated sequence trees for 14 diploid _Crataegus_ accessions (B; Table 1) and for the same 14 diploids plus 10 related allotetraploid _Crataegus_ accessions (C; Table 1). Red labels in (A) represent the nodes in (B); black labels in (A) represent the nodes in (C). Unless indicated otherwise (italicized proportions), nodes in (B) have bootstrap support = 1.00 or (C) ≥ 0.98. Scale bars for branch lengths are in substitution units. Pie charts show gene (gCF, red) and site (sCF, yellow) concordance values for nodes, as percentages. Green rectangles denote nodes at which the gene discordances (gDF$_1$, gDF$_2$) both exceed 5%. Orange rectangles denote nodes at which the chi-squared test rejects incomplete lineage sorting as the underlying cause of discordance among gene trees and sites (_Lanfear, 2018_). Compare Table S2. Accessions are coded as in earlier figures by _Crataegus_ subgenera (_MESP, BREV, CRAT, AMER, SANG_) and sections (color; _Mesp, Brev, Crat, Cocc, Macr, Doug, Sali_, and series **_Sang_**_uineae_ and **_Nigr_**_ae_ in section _Sanguineae_; Table 1). All trees rooted using the corresponding sequences from the genome of apple, _Malus ×domestica_ (_Velasco et al., 2010_).

in the combined sample (Fig. 5A; Table S4). The effect of the adding the tetraploids on the proportion of sites supporting individual nodes is much less marked, from 30–65% to 30–55% (Fig. 5A). With both types of concordance values there is a generally linear trend to the effect of adding the tetraploid accessions. In the case of the gCF values, however, four nodes show lower values than this trend would predict: 29 (monophyly of subgenus _Americanae_), 37 (monophyly of subgenus _Sanguineae_), 42 (monophyly of the two diploid _C. suksdorfii_), and 47 (monophyly of the two diploid _C. saligna_). The first two departures could be attributed to the multiplicity of topologies stemming from tetraploid individuals
sharing sequences with individuals in the other subgenus. The latter two are probably linked in a similar way to diploids joining other clades because they share sequences with their allotetraploid descendants. As discussed by *Lanfear (2018)*, the bootstrap support values for these nodes are 90–100%, reflecting the large sample size (average total sequence length 424.5 kbp per accession) and its effect on the sampling variances for these nodes. Only two of the internal nodes in the *Americanae* clade, and four in the *Sanguineae* one, have bootstrap values below 83% (Fig. 5C, italicized support values).

Following *Lanfear (2018)*, further examination of the gene and site concordance values makes possible a test of $H_0$: ILS is responsible for approximately equal numbers of genes supporting alternative topologies. With our data, ILS is rejected for only two nodes on the diploids-only IQ-TREE: the one supporting the two *C. suksdorfii* accessions, and the one supporting the two accessions belonging to *C.* sect. *Sanguineae* (nodes 25 and 27 in Fig. 5B). On the diploids + tetraploids tree (Fig. 5C), in addition to node 43 (sect. *Sanguineae*) $H_0$ is rejected for nodes 33 (*C. crus-galli*, *C. punctata*, and 4*x C. macracantha*, s11; Table 1) and 40 (three 4*x C. douglasii*). These represent, respectively, the relative homogeneity of the sequences in the two *Sanguineae* accessions sampled, the heterogeneity of relationships between three accessions in undersampled subgenus *Americanae*, and a similar heterogeneity in topological relationships between three geographically disparate samples of allotetraploid *C. douglasii*. In other words, ruling out ILS leaves room for more than one possible explanation of relationships above a given node.

Comparing the ASTRAL-III coalescent trees (Fig. 3) with the IQ-TREE trees for the concatenated sequence data (Fig. 5), we note that the diploids-only coalescent tree (Fig. 3A) does not include the two section *Sanguineae* accessions in a single clade, in contrast to the diploids+tetraploids tree (Fig. 3C) and the ML trees (Fig. 5). This could perhaps be explained as a consequence of undersampling the complexity of subgenus *Sanguineae*. In both sets of multilocus trees support levels for the branches within the subg. *Americanae* clade are low (Fig. 3, local posterior probabilities < 0.95; Fig. 5, markedly low gCF values).

## DISCUSSION

Our study draws attention to the diversity of tree topologies encompassed by nuclear gene trees estimated first for a sample of 14 diploid-only *Crataegus* (hawthorn) accessions, and then for the same diploid accessions and 10 related tetraploid accessions. Incongruence of this kind is proving to be ubiquitous in phylogenomic studies (for early examples, see *Leigh et al., 2011*; *Rokas et al., 2003*). The 245 gene trees for the diploid-only sample (244 for the mixed ploidy sample) represent the nuclear low copy number loci with complete data out of a sample of 257 loci in all obtained by target capture sequencing (*Weitemier et al., 2014*). For each of the same two sets of accessions we also obtained a plastome tree. The topologies of both these trees reflect the relationships between *Crataegus* subgenera seen in earlier molecular work (*Lo et al., 2009a*; *Zarrei et al., 2015*), as well as the morphological and biogeographic relationships between the subgenera (*Ufimov & Dickinson, 2020*), and so we used them as reference trees against which we compare the single (and multi-) locus nuclear trees. In this way we seek to corroborate earlier evidence for gene flow within and

between subgenera (*Lo, Stefanović & Dickinson, 2009b*; *Lo, Stefanović & Dickinson, 2010*; *Zarrei, Stefanović & Dickinson, 2014*), as distinct from the effects of incomplete lineage sorting (ILS; *Gitzendanner et al., 2018*; *Li et al., 2018*; *Lloyd Evans, Joshi & Wang, 2019*; *Magdy et al., 2019*) and possible artifacts of the sequence assembly process.

Despite the diversity of individual nuclear gene tree topologies (Fig. 4), our nuclear multilocus trees (Fig. 3) depict phylogenies very similar to our plastome trees (Fig. 2) in which the two most heavily sampled subgenera (*Americanae*, *Sanguineae*) are sister to each other, and in which their relationships with the other *Crataegus* subgenera are much as seen in earlier work (*Lo et al., 2009a*; *Zarrei et al., 2015*). Consistent with these previous studies, hybridization between subgenus *Americanae* and subgenus *Sanguineae* was documented for the origin of *Sanguineae* tetraploids, but not for a tetraploid *Americanae* species. Examination of the gCF and sCF values explicitly rejected a role for ILS for only a very small number of nodes. The multispecies coalescent model explicitly accounts for ILS (*Rabiee, Sayyari & Mirarab, 2019*; *Zhang et al., 2018*), and for this reason we consider these trees (Fig. 3) to be the most accurate estimate of phylogeny. However, the support for inter-subgeneric hybridization obtained in the HyDe analysis (Table 3) and the increased topological diversity of the diploids + tetraploids trees (Fig. 4) indicate that ILS alone cannot explain all gene tree heterogeneity. We note that our use of plastome trees as references against which to compare our trees derived from nuclear loci follows a widely accepted paradigm for discovering the occurrence of hybridization (*Gitzendanner et al., 2018*).

## Phylogeny

Our results attest to the way in which robust multilocus phylogenies can be obtained from large samples of topologically diverse gene trees. Likewise, these multilocus trees are congruent with the morphology- and geography-based infrageneric classification of *Crataegus* (Table 1; *Ufimov & Dickinson, 2020*) supported up to now by less comprehensive molecular datasets (*Lo & Donoghue, 2012*; *Lo et al., 2009a*; *Lo, Stefanović & Dickinson, 2007*; *Lo, Stefanović & Dickinson, 2009b*; *Zarrei, Stefanović & Dickinson, 2014*; *Zarrei et al., 2015*). In addition to our results, two recent studies of Chinese *Crataegus* (*Hu et al., 2021*; *Wu et al., 2021*) further demonstrate the utility of plastome phylogenies in resolving closely related species of the genus.

## Comparison of diploids-only gene trees

We discuss our results first with respect to the single-locus trees, the coalescent and the concatenated multilocus trees, and the plastome tree built using the diploids-only sample. The reference trees (plastome tree, Fig. 2A, coalescent tree, Fig. 3A, and concatenated tree, Fig. 5B) are highly congruent with each other in distinguishing subgenera *Americanae* and *Sanguineae*, but differ in topological details within these clades. They also differ in which of the early-arising subgenus-level groups (*Brevispinae*, *Crataegus*, *Mespilus*) is sister to the rest of the genus. However, only 19 of the single-locus trees (Fig. S1) show a high degree of concordance (here, the related tips concordance, rtCF; *Kendall, Eldholm & Colijn, 2018*), exceeding the third quartile by more than $1.5\times$ the interquartile range of the concordance values (documentation or the **boxplot** function; *R Core Team, 2016*).

The incongruence of the majority of the single-locus trees (Figs. 4A, 4C, 4E) is unlikely to be due to hybridization. Homoploid hybrids between diploid hawthorns are uncommon, and are best known for involving *C. monogyna* Jacq. and other species with which it is sympatric either in its native range or where it has been introduced (*Christensen, 1992*; *Christensen et al., 2014*; *Phipps, 2005*; *Phipps, 2015*). Only three taxa in the sample studied by *Lo et al. (2009a)* exhibited incongruence between trees built using chloroplast loci and ones built from nuclear ones, and none of these (putative trans-Atlantic paleohybrids) is included in our samples here. The variation in the diploids-only sample single-locus gene tree topology (Figs. 4A, 4C, 4E) therefore probably arises from unknown proportions of ILS, paralogy, recombination, and the potential presence of misassembled chimeric sequences.

## Comparison of diploids + tetraploids gene trees

The relatively greater dispersion of the diploids + tetraploids trees (Figs. 4B, 4D, 4F) evidently represents the effect of hybridization added to those of ILS and the other processes mentioned above. With diploids + tetraploids sample the reference trees (plastome tree, Fig. 2C, and multilocus tree, Fig. 3C) are also highly congruent in both overall topology (resembling the corresponding diploids-only trees) and in the placement of the *C.* subg. *Sanguineae* tetraploids with their inferred maternal parents (Fig. 3B). In the diploids + tetraploids single-locus trees corresponding to the 19 diploids-only trees that are highly congruent with the reference trees (Fig. S1) the frequent placement of the *Sanguineae* allotetraploids in the *C.* subg. *Americanae* clade suggests that sequencing and assembly procedures have captured a sequence derived from the *Americanae* parent of the allotetraploid. The strong signal obtained with the admixture coefficients ($\hat{\gamma}$; Table 3; Fig. S3) also supports hybridization as an explanation of gene tree incongruence, as does the morphological intermediacy of the allotetraploids (Table 2).

In the diploids + tetraploids sample, the incongruence of the remaining 225 single-locus trees likely does result from the presence of the tetraploids (*Sanguineae* allotetraploids and their putative *Americanae* parents) together with ILS and the other processes mentioned above. As described for the diploids-only sample, the gene and site concordance and discordance values for the diploids + tetraploids multilocus tree (Fig. 5C) provide further insights. Nodes supporting clades affected by the addition of the tetraploids (nodes 29 and 37, Fig. 5C) have much reduced gCF support, suggesting that these branches as they appear in the multilocus tree are in fact now supported by many fewer individual gene trees (Fig. 5A).

## Multivariate comparisons of gene trees

As others have recently found (*Amenta & Klingner, 2002*; *Bogdanowicz, Giaro & Wróbel, 2012*; *De Vienne, Ollier & Aguileta, 2012*; *Gonçalves et al., 2019*; *Huang & Li, 2013*; *Huang et al., 2016*; *Jombart et al., 2017*; *Kendall, Eldholm & Colijn, 2018*; *Richards et al., 2018*), comparisons of the topologies of large numbers of gene (or other) trees is facilitated by graphical methods like principal coordinates analyses of distances depicting the resemblances of these trees with respect to cladistic relationships between the objects

of study. Principal Coordinates Analysis (PCoA) is a well-known and commonly employed ordination method based on eigenanalysis of a transformed resemblance matrix (*Gower, 1966*). Methods like PCoA (and Principal Components Analysis) find successive orthogonal axes corresponding the directions in which variation in a multidimensional sample is greatest (*Legendre & Legendre, 1998*). For a given sample (such as either the diploid-only or diploids+tetraploids sample), some resemblance functions capture this variation in fewer such axes. Here, because our sample is taxonomically structured and the RT distance incorporates this structure (subgenera, sections), more of the variation in our two samples is captured in fewer dimensions than is the case with the distances that ignore taxonomic structure (MV, SMD). We have demonstrated the value of a phenetic approach to tree comparisons, calculating Euclidean distances between trees from three descriptors of their structure (*Podani, 1982*), and using a more recent distance function that takes into account relationships extrinsic to the trees themselves (*Kendall, Eldholm & Colijn, 2018*). These relationships (subgenera, sections of *Crataegus*) in our opinion do not contribute circularity, but rather enable us to better discern the relationships between tree topology and taxonomic structure relevant to our enquiry into the occurrence of hybridization. In contrast, the SMD, PMD, and CMD (individually or combined) are entirely agnostic with respect to these relationships. We found that PCoA of the related tree distances gave us the lowest dimensional summary of the diversity of tree topologies in both our diploids-only and diploids + tetraploids samples (RT-PCoA, Figs. 4A, 4B; Fig. S4; *Legendre & Legendre, 1998*). Nevertheless, the first two dimensions of all three PCoAs (Fig. 4; Fig. S4) showed the same contrast in topological diversity between the diploids-only and diploids + tetraploids samples that we interpret as the effect, primarily, of ILS and ILS plus hybridization, respectively. We suggest that others will also find this phenetic approach useful as it generates its own insights into variation in tree structure and complements those to be gained from IQ-TREE where studies generate large numbers of gene trees.

## Taxonomic, evolutionary, and biogeographic implications

Our results provide support for the infrageneric classification of *Crataegus* at the levels of subgenera and sections, as used here (Table 1; *Ufimov & Dickinson, 2020*). In response to the comments by *Phipps (2016)* on earlier molecular results (*Lo et al., 2009a*; *Lo, Stefanović & Dickinson, 2007*; *Zarrei et al., 2015*), we note first that the monophyly of the clade *Crataegus + Mespilus* is well-established by studies showing that a *Hesperomeles + Crataegus + Mespilus* clade (*Li et al., 2012*) is sister to an *Amelanchier* clade, and that this combined clade, in turn, is sister to most or all of the remaining Malinae (*Liu et al., 2020*; *Lo & Donoghue, 2012*). Second, we observe that our sample of gene trees (Figs. S1, S2; compare Figs. 3B, 3D) and our plastome trees (Figs. 2B, 2D) provide support for both excluding or including the medlar in *Crataegus*. This ambivalence suggests to us that radiation of hawthorns and medlars (or, of subgenera within *Crataegus*) occurred relatively rapidly (much as with the genera of the Maleae; *Campbell et al., 2007*; *Lo et al., 2009a*), with only a single species of medlar and *C. brachyacantha* persisting to the present, representative of their respective subgenera. We note wide acceptance of the idea that taxonomic rank above the species level is to a high degree arbitrary (*Stevens, 1997*), so that given the morphological similarities
between *C. germanica* and some, if not all, of the remaining *Crataegus* species we see no compelling reason to maintain two separate genera (*Ufimov & Dickinson, 2020*).

## CONCLUSIONS

We have demonstrated that the topological diversity of individual low copy number nuclear gene trees that we obtained by Next Generation Sequencing can nevertheless produce well-supported multilocus phylogenies, in this case for the genus *Crataegus*. This result, in our diploids-only sample (in which evidence for hybridization is scant) suggests an important role for ILS. However, admixture statistics suggest that the increased diversity attendant on including *C.* subg. *Sanguineae* tetraploids and their probable *Americanae* pollen parents together with the same diploids is due to the hybrid origin of the *Sanguineae* tetraploids, even if there are also effects of additional ILS in the larger sample.

These results are consistent with phylogenomic studies of other plant genera where incongruence among numerous nuclear loci has been attributed to a combination of ILS and interspecific hybridization and introgression (*Bernhardt et al., 2020*; *Carter et al., 2019*; *Karimi et al., 2019*; *Morales-Briones, Liston & Tank, 2018*; *Murphy et al., 2020*). In our study and these others, the integration of species tree approaches that account for ILS, measures of gene tree discordance, and network analyses have been effectively employed to tease apart the relative contributions of these processes.

We note too that the multilocus phylogenies are highly congruent with ones we obtained from whole plastome sequences, and with a morphology-supported subgeneric classification of the genus. Our results, based on vastly more sequence data than has been available previously, support earlier ones suggesting that intersubgeneric hybridization, aided by the occurrence in *Crataegus* of gametophytic apomixis, has played a much more important role in *Crataegus* evolution than has been previously recognized. This result parallels recent observations on other large genera of Rosaceae subtribe Malinae, such as *Amelanchier* and *Sorbus*. Our conclusions concerning the roles of ILS and hybridization in *Crataegus* arise notably from our use of tree-tree comparisons and in particular from our use of a metric that employs information relevant to our enquiry, namely the infrageneric and sectional affiliations of our samples.

*Liston (2014)* designed probes for targeting 257 nuclear loci in Rosaceae, using apple, peach and strawberry, the three genome sequences available at that time. This is the first application of the apple probes, and the 95% success rate in locus assembly demonstrates their utility. The fact that *Crataegus* and *Malus* belong to different clades of the Malinae (*Liu et al., 2020*; *Lo & Donoghue, 2012*) suggests that these probes will be very effective across the entire apple subtribe. An increasing number of studies are relying on the "universal" for flowering plants Angiosperm-353 set of probes (*Johnson et al., 2019*) suggesting that clade-specific probes may not be necessary. However, it is feasible to combine both universal and clade-specific probes in a single study (*Larridon et al., 2020*; *Shah et al., 2021*; *Siniscalchi et al., 2021*), and this has the benefit of further increasing the number of loci available for phylogenomic comparison.

## ACKNOWLEDGEMENTS

TAD is indebted to the Arnold Arboretum of Harvard University, Jamaica Plain, Massachusetts; the Jardin Botanique de Montréal, Québec; Trey Lewis, De Soto Parish, Louisiana; and the University of California Botanical Garden at Berkeley for permission to collect *Crataegus* species on their properties. TAD is likewise grateful to the staff, volunteers, and (over many years) the students and other assistants at the Green Plant Herbarium (TRT) of the Royal Ontario Museum. Many of the collections could not have been made without the help of field guidance from Rebecca Dotterer, Ron Lance, Rhoda M. Love, James B. Phipps, and the late Steve Brunsfeld. Finally, TAD thanks the collectors listed in Table 1, together with David Baxter, Adam Dickinson, John Dickinson, Rodger Evans, Fannie Gervais, Graeme Hirst, Hazrah Moothoo, Sophie Nguyen, Tamra Prior, Cassandra Shaw, and Mehdi Zarrei for their assistance in the field. TRT's collection of *Crataegus* leaf tissue preserved with desiccating silica gel was curated by Mehdi Zarrei with the assistance of Ionatan Waisgluss. AL thanks Mark Dasenko and Brent Kronmiller of the Oregon State University Center for Genome Research and Biocomputing for sequencing and preliminary bioinformatic analysis, respectively.

### Funding

This research was made possible with the support of the Royal Ontario Museum (ROM) DMV Acquisition & Research Fund, 2013; Discovery Grant (A3430; re-applied for at 3–5 year intervals), Natural Sciences and Engineering Research Council of Canada, 1987–2015; Royal Ontario Museum departmental fieldwork funds (Natural History, Center for Biodiversity & Conservation Biology, Science, Science Cooperative Field Studies), 1988-2008; Research Grants, ROM/ROMCA Special Research Fund, 2014–2019; and Strategic Project Grant (381073), Natural Sciences and Engineering Research Council of Canada, 2010–2012, joint with Prof. S. Stefanović (University of Toronto – Mississauga), Paul Shipley (University of British Columbia – Okanagan), S. Proctor (University of Alberta) and the Naturally Grown Herb and Spice Growers Co-operative (HerbPro; Edgewood BC, J. Lee, President); Paula Brown (British Columbia Institute of Technology) materially assisted us in obtaining this grant. Student assistants in the ROM Green Plant Herbarium were co-funded by the University of Toronto Work Study program; all of the University of Toronto and NSERCC funding was facilitated by the Department of Ecology and Evolution (and its predecessor, the Department of Botany). The funders had no role in study design, data collection and analysis, decision to publish, or preparation of the manuscript.

### Grant Disclosures

The following grant information was disclosed by the authors:
Royal Ontario Museum (ROM) DMV Acquisition & Research Fund, 2013.
Discovery Grant: A3430.
Natural Sciences and Engineering Research Council of Canada, 1987–2015.

Royal Ontario Museum departmental fieldwork funds (Natural History, Center for Biodiversity & Conservation Biology, Science, Science Cooperative Field Studies), 1988–2008.
Research Grants.
ROM/ROMCA Special Research Fund, 2014–2019.
Strategic Project Grant: 381073.
Natural Sciences and Engineering Research Council of Canada, 2010–2012.
University of Toronto – Mississauga.
University of British Columbia – Okanagan.
(University of Alberta) and the Naturally Grown Herb and Spice Growers Co-operative.
British Columbia Institute of Technology.
Student assistants in the ROM Green Plant Herbarium.
The University of Toronto Work Study program.
The Department of Ecology and Evolution.

## Competing Interests

The authors declare there are no competing interests.

## Author Contributions

- Aaron Liston conceived and designed the experiments, prepared figures and/or tables, authored or reviewed drafts of the paper, and approved the final draft.
- Kevin A. Weitemier performed the experiments, analyzed the data, prepared figures and/or tables, and approved the final draft.
- Lucas Letelier, Yu Zong and Lang Liu analyzed the data, prepared figures and/or tables, and approved the final draft.
- János Podani analyzed the data, authored or reviewed drafts of the paper, and approved the final draft.
- Timothy A. Dickinson conceived and designed the experiments, analyzed the data, prepared figures and/or tables, authored or reviewed drafts of the paper, and approved the final draft.

## Field Study Permissions

The following information was supplied relating to field study approvals (i.e., approving body and any reference numbers):

The Arnold Arboretum of Harvard University, Jamaica Plain, Massachusetts; the Jardin Botanique de Montréal, Québec; Trey Lewis, De Soto Parish, Louisiana; and the University of California Botanical Garden at Berkeley gave permission to collect Crataegus species on their properties.

## DNA Deposition

The following information was supplied regarding the deposition of DNA sequences:

The data is available at NCBI SRA BioProject PRJNA673921.

## Data Availability

Assembled sequence alignments and phylogenetic trees for the plastomes and nuclear loci are available at Zenodo: Liston, Aaron, & Dickinson, Timothy A. (2021). Phylogeny of Crataegus (Rosaceae) based on 257 nuclear loci and chloroplast genomes: evaluating the impact of hybridization [Data set]. Zenodo. https://doi.org/10.5281/zenodo.4209576.

## Supplemental Information

Supplemental information for this article can be found online at http://dx.doi.org/10.7717/peerj.12418#supplemental-information.

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
