# Peer review of "Phylogeny of Crataegus (Rosaceae) based on 257 nuclear loci and chloroplast genomes: evaluating the impact of hybridization"

_PeerJ, doi:10.7717/peerj.12418_

## Round 0.1 · original submission · Major Revisions

Both reviewers and I found the paper adequate and important for PeerJ. Please consider every issue raised by the two reviewers, both included a number of comments and one of the reviews is outstanding. I look forward to seeing these changes.

Reviewer 1 ·

Basic reporting

• The article is well written in good professional English, although I find it a bit too (unnecessary) technical in some of the sections.
• Literature references are sufficient and the context is well provided.
• The structure is ok, but there is a bit too much discussion in the results section.
• The figures I have seen in the pdf, have too low resolution. They are not publishable.
• In my view, one figure should be removed, as well as one of the tables. The figures need work and I have suggestions below.
• I have seen no indication in the manuscript that the data (as analysed) are publicly available.
• The manuscript is weak on its aims and the relevance of its analytical methods, but contains proper sections on results, discussion, and conclusions.

Experimental design

• I consider the manuscript to be within the stated aims and scope of the journal.
• The manuscript presents the background well, but not its aims.
• The methods used are presented in much detail, but not what they are expected to show.
• The authors do not present clear reasons for using some of the analytical methods.
• The study (is stated to) supplement earlier results by adding much more data of a new kind - and it does.
• The methods of analysis are partly described in sufficient detail, but not all of them. Some are not possible to replicate from the manuscript information (see below)

Regarding field study permits
I was specifically asked by the journal to check the author’s field study permits. I find this is not relevant. All accessions used (listed in table 1) were either previously published elsewhere or the collections made by collectors who are not authors.

Validity of the findings

No comment

Additional comments

In general, I consider the lack of clarity of the aims of the manuscript, and the methods to be its weaknesses. In simple terms: I want to understand why the study was performed, why these particular methods where used, and what they might be expected to show (or not) – and I do not find this clearly stated.

Judging from the title, the idea is to acquire 257 nuclear loci and most of the plastid genome, and use those data to evaluate the impact of hybridization in Crataegus. However, this is not quite clear from the text itself. The authors state what they have done, but not what the aims are (“Here we supplement the earlier results”), nor how the methods chosen were expected to bear upon the problem at hand.
It occurs to me, however, that if the problem was to test the hypotheses of hybrid origin of certain North American species, why not use phylogeny directly? Instead, some indirect statistical methods are used. The nuclear loci used were previously selected because they were putatively orthologous (ms line 134) in a wide range of Rosaceae species. This would mean that a relatively recent tetraploid could be expected to have two copies of each of these loci. The position of these in a phylogenetic analysis including both copies might indicate the origin of the putative hybrid. As far as I know, such a direct approach would have been novel in Crataegus, and might have given more direct results regarding hybrid origins.
The authors were using HybPiper (ms line 150), but apparently not its extension HybPhaser which was made for cases like this (https://www.biorxiv.org/content/10.1101/2020.10.27.354589v1). It is recent but available (https://github.com/LarsNauheimer/HybPhaser) and the manuscript should at least include a discussion as to why such a more direct phylogenetic approach was not considered. Although HybPhaser is new, the idea of phasing target capture datasets is not, and other pipelines have been published (e.g. Andermann et al. 2019). Instead, in this study, information that might bear on the hybrid origin of species was apparently discarded: “all ambiguous bases replaced with Ns” (ms line 152). (It is not quite clear to me from the manuscript, however, if this was only done in the six loci flagged as potential paralogs by HybPiper. This needs to be made more clear.)

There are unexplained inconsistencies in the methods that I think should be explained. For the ML estimations of the nuclear target capture data, the authors used IQ-TREE, but RAxML for the plastome data. Why? The authors of IQ-TREE claim that their program yields results with a better likelihood than does RAxML (and PhyML). Would that not apply to plastome data? The results were compared, and perhaps inconsistencies between plastome and nuclear data might be partly explained by using different softwares to estimate the phylogenies?

Regarding the alignments, it is stated that MAFFT was used (ms lines 154, 202). Although the version number was indicated, all other parameter settings were omitted and they may be of great interest to someone who wants to repeat the analyses. The developers of the MAFFT software have even simplified the settings of groups of parameters by including helpful short commands which might be cited (such as “linsi”), simplifying this. Also, since the alignments (both nuclear and plastome) are said to have been manually edited, the end products used for the analyses in the manuscript should have been made available in an open repository. I see no indication of this in the manuscript.

The authors used Astral-III to estimate species trees from multilocus data. As far as I know, Astral takes into account ILS – but not allopolyploidy. Still, the polyploid species/samples (that in several places in the manuscript are said to be allopolyploids: lines 113, 234, 243, 253, 388, 438, figure legends, table 2 etc) were included. I do not know what the possible effects are of including data that violate the models of the program in such a way, but since the authors use Astral-III under the circumstances, a discussion of this methodological problem is necessary at least (see below).

It is unclear what the “Neighbor-net” in fig. 1 is for. It is mentioned in the introduction, in the materials and methods section, but not in the results and whatever it shows (unclear) is never discussed.

The congruence of the gene trees to each other and to the previous classification was investigated, by using a number of (statistical) methods. It is difficult to evaluate this section because it it not very clear (to me) why precisely these methods were used and not any others – or what results might be expected. For example, regarding the calculation of tree distances, the authors say (ms line 287):
“We calculated distances between our trees in order to display the variation they exhibit in the low-dimensional spaces of the first Principal Coordinates Analysis (PCoA) axes for which the proportion of variance accounted for by the axis exceeds that expected under the broken-stick model”. It would be helpful to get an explanation of what these calculated distances (as well as the other tree descriptors) are supposed to mean in relation to the aims of the paper.

The tree collapsing algorithm used (see comment to figure 2) seems not to always work well for the purpose it is being used when comparing the separate gene trees with the previous classification. This may constitute a problem for the interpretation of those comparisons.

Results

Plastome trees

Sweeping statements like “trees were generally well-supported” are not helpful, and thus should be avoided when in the diploids+polyploids tree (fig. 2D) almost half of the nodes have low support (marked by an asterisk in fig. 2D).

Gene trees

Most of the work on the gene trees focused on sorting and checking them with reference to previous classification (and the plastome tree). Apparently, this led to the selection of only 19 trees (out of 245) that showed “markedly high rtCF values” (ms line 368) which possibly indicates that they agreed well with the tree implied by the classification – although this is not made very clear in the results. This small sample of select trees (less than 8% of the obtained loci) were then further acted upon and some compared in detail. What was the rationale for this? Are they more “true” than the others so that the details are more interesting?

Phylogenetic invariants

The analyses of the species mentioned strongly suggests that they are intersubgeneric hybrids, apparently by “vanishing small p-values”. This description should be made clearer, because in most cases hypotheses are rejected when p-values are small. To understand, the reader needs to know what was being tested. It is not made clear in Materials and methods, nor in Table 3.

Multilocus trees

If there were allopolyploids in the analyses, is it not worrying that the coalescent trees where “highly concordant with the plastome trees”? In particular, if the analysis of phylogenetic invariants showed intersubgeneric allopolyploids to be present, would a strong agreement with the plastome tree be expected? I may be wrong, and I may have erroneously thought that “concordant” had to do with “congruent”.

In addition to the differences reported between that plastome tree(s) and the multilocus tree(s), “SANG Sang wilsonii s02” is resolved differently. In both plastome trees it is well supported in a clade with “SANG Nigr nigra s23”, while in the multilocus diploids only tree, it is well supported as sister to the other “SANG” samples. This is finally discussed at the end of the results section. There is actually a fair amount of discussion towards the end of the results section (line 435 and onwards) parts of which might be better to move to the discussion section.

Discussion

In the paragraph on lines 485–503, the causes of the tree heterogeneity is discussed. Although ILS was not rejected, it seems that hybridisation is the favoured cause (“all suggest that ILS has not been more important that hybridization”). I do not disagree with this conclusion, but the discussion would benefit from a more in-depth discussion of why the ILS test fails to reject ILS here, and which of the tests performed here might be of more general use. After all, as pointed out in the beginning of the discussion, results similar to these are not uncommon in phylogenomic studies. It would also be of interest to include a discussion of how such problems have been dealt with previously.
In this paragraph, it is of course a bit contradictory to (1) note that ILS is not rejected, (2) take that as support for considering the multilocus coalescent tree to be the most accurate estimate of the phylogeny, and (3) suggest that ILS has been less important here than hybridisation.

To reiterate a general comment above about the multilocus coalescence methods: Here it would be appropriate to discuss the possible effects when including allopolyploids in analyses such as performed with Astral-III, that possibly violate the models of the program. This is particularly interesting among plants where allopolyploidy is common. Perhaps it is not a problem, and the resulting tree is still the best estimate of phylogeny?

I still find the discussion about the Multivariate comparisons of gene trees obscure. This may be because I am not well enough versed in Principal Coordinates Analyses speak, but I would have liked to see something more understandable than (line 568) “We found that the related tree distances gave us the lowest dimensional summary of the diversity of tree topologies”, and the like.

Regarding the discussion on the position and ranking of the medlar, it is notable that although the plastome data (not surprisingly) yields a position in Crataegus for it like previous plastid data, the nuclear data (fig. 3, fig. 5) does not! What needs to be pointed out here is that since the apple was used as outgroup, and nothing else was sampled outside Crataegus, the nuclear data in this study has no bearing on the position of the medlar except that it is outside of Crataegus in the strict sense. If more taxa had been included, Mespilus need not have been sister to Crataegus, even.


Figures

All figures are of very low resolution and they contain visible artefacts from too much jpeg compression. They can not be published in this form.

Figure 1. The figure seems superfluous, except as general information. In the text it is only referred to in the introduction. It is not really used for anything and not reported as a result or discussed. In the Materials and methods section it is referred to as “Phylogenetic network”, which I think is incorrect.
The legend incorrectly claims the “tree” to be rooted. No, this shows an unrooted tree. Indeed, one of the branches connects the apple to the rest, but this does not mean that the tree is rooted.
The legend indicates the loci to be “low copy number loci”. Clearly, only a single copy is used here, and elsewhere “putatively orthologous” is used.
The description of labels is complex and unclear, and refers (sample numbers) to table 1 where they are hard to find. Similar (but not identical) labels are found in table 3. For easy comparison they should be identical, not just similar. I suggest that the authors use the same labels throughout.

Figure 2. The figure shows four trees. Two of these are relevant (A, D) while the two simplified are not (B, C), because the two former includes all information in the two latter. Also, B and C are confusing as they have been ladderized differently (outgroup at bottom instead of at top), and incorrect because they resolve nodes that are not supported (the node BREV+MESP+AMER+SANG in B, the node Brev+Mesp+Macr+Cocc+Doug+Sang+Sali in C, and incorrectly indicating Macr+Cocc to be sisters in C). I doubt that the collapsing algorithm used actually works correctly for the purpose it is used in this manuscript.
If B and C are retained, taxa need to be organised in the same way as A and D with outgroup to the same side and clades ordered in the same way. I am aware that this does not change the relationships in any way, but it should be done for ALL figures so that they can be more easily compared by the reader.

Figure 3. Here, for some reason, the figure order is different from figure 2: A and B shows the full trees, while C and D show the simplified trees. Like in figure 2, the simplified trees are unnecessary as well as confusing, being differently ladderized. Here also, tree A and B are drawn with the outermost node collapsed, while C and D are drawn resolved. Furthermore, the AMER and SAMG nodes have been rotated in tree B which is confusing. It should be like tree A (and trees A and D in figure 2) to increase clarity.
The last sentence of the legend distinctly looks like an editing mistake.

Figure 4. I am not sure if this figure shows more than the rather obvious “if you add more taxa the trees will change”. The legend is long, very complex and hard to understand due to lots of abbreviations. Although long, the legends on the axes are still not explained (such as RT-PCo2 or PCo1-PCo28).
The legend above the scattergrams say “246 Diploid-only trees” and “245 Diploid+tetraploid trees”. Elsewhere, the corresponding numbers are 245 and 244, respectively. I assume that it is incorrect here?
The last sentence in the legend mentions that the trees were rooted before calculating distances. Why was this necessary? In which way would rooting affect the distance?

Figure 5. This is a very complex figure and it is so full of graphical elements that it is difficult to see the branching pattern. The word “cluttered” comes to mind, and I suggest that the authors consider not trying to show all of this in one figure.
In A and B, the clades AMER and SANG are rotated as compared to figure 2. They should be in the same order for easy comparison.

Table 1.
Is “stamen number per flower” relevant here? It is only shown in tables 1 and 2 and not mentioned or discussed in the text.

Table 2.
This table seems unnecessary. It is only cited once in the Materials and methods section and the information in it is not used or discussed.

Table 3. Legend is cut off.
It would be helpful to include a text on what H0 and H1 represents, not just that the “admixture statistic” is zero or larger than zero. So if the p value is low enough to reject H0, what does it mean? That the hybridization hypothesis is favoured? If this is the case, it would help the reader to say this.
The p-value for four hybrids is 0.0 in the table. Does this mean exactly zero? Could it mean 0.0001? I am unfamiliar with the usage.
The labels for samples here should be identical to those used elsewhere (such as in figure 1 etc). Here it is “s12riv” while in figure 1 it is “s12_riv”. I can guess that they are the same, but why make them different?

·

Basic reporting

This is an important contribution from an outstanding team of researchers. The work is technically sound, the methods are rigorous, and the interpretations are sound. That said, I found the writing in many places to be quite dense and not accessible and engaging for readers outside of other specialists in Crataegus and/or the tree comparison methods. I understand that PeerJ does not consider possible readership an important criterion for publication; nonetheless, there are some aspects of the presentation that I think could be improved for clarity and precision in any case, as follows:
1. Lines 50-52: The last sentence of Abstract-Results (“Despite the high heterogeneity of individual gene trees, we corroborate earlier evidence for the importance of hybridization in the evolution of Crataegus.”) is quite general and not very informative, even for the Abstract. A more detailed and specific summary of the key findings regarding hybridization in the genus would be helpful.
2. Line 55: The construction “hybridization…has been contentious” is awkward; the question of the importance of hybridization has been contentious, not the process itself.
3. Line 61: The phrase “relatively early” is vague.
4. Lines 62-63: Use of present tense seems incongruous.
5. Lines 69-73: More detailed explanation of what past studies have shown, and how, would be helpful (though not necessary for those already familiar with these works).
6. Lines 79-81: Again, I think more specific details, especially examples of how the processes mentioned have contributed to “taxonomic complexity” would be helpful, but specialists may not find this necessary.
7. Lines 89-99: Perhaps I’m stuck on tradition, but I found the description of some details of methods and the citation of Figures depicting specific results of the study in the Introduction jarring.
8. Lines 556-576: This paragraph is very dense and focused on details of the results, rather than more general conclusions. It reads more like Results than Discussion.
9. Lines 578-607: Most of this paragraph concerns the question of whether or not Mespilus should be considered as a distinct genus from Crataegus, a subjective taxonomic decision given that the phylogenetic results are consistent with either treatment. The amount of space devoted to this one question that was not presented as a central focus of the study seems disproportionate.
10. Lines 611-630: Even in the Conclusions section, the descriptions are primarily focused on details of the Results rather than on big-picture “take-home” messages. On the other hand, the final sentence, especially the last phrase (“unforeseen consequences of methodological choices”) is so vague as to not be very informative.

Experimental design

No comment.

Validity of the findings

No comment.

---

## Round 0.2 · Minor Revisions

I appreciate your effort for considering the issues raised previously by the two reviewers. However, the decision of minor changes is based in a concern which I share with one of the reviewers, in that justification is needed to understand the use of two different (IQ-TREE and RaxML) programs for constructing trees for the plastid and nuclear datasets.

Reviewer 1 ·

Basic reporting

No comment

Experimental design

In my view the sequence alignment methods, in particular when using MAFFT, are not described in sufficient detail for replication. Using "default" settings and "auto" selection of algorithm is unfortunately a common practice, but it might make analyses difficult to replicate if default settings or method selection algorithms change in the future. I do not think this is a major problem in this case, however, as concatenated aligned data sets have been submitted to a repository.

Validity of the findings

No comment

---

## Round 0.3 · accepted · Accept

Thank you very much for standardizing analyses performed with the plastome and the nuclear data matrices. Also for adding a recent reference and changing some issues. I appreciate very much your effort.